# An inner activation gate controls TMEM16F phospholipid scrambling

Trieu Le[1], Zhiguang Jia[2], Son C. Le [1], Yang Zhang[1], Jianhan Chen [2,3] & Huanghe Yang [1,4]

Transmembrane protein 16F (TMEM16F) is an enigmatic $Ca^{2+}$-activated phospholipid scramblase (CaPLSase) that passively transports phospholipids down their chemical gradients and mediates blood coagulation, bone development and viral infection. Despite recent advances in the structure and function understanding of TMEM16 proteins, how mammalian TMEM16 CaPLSases open and close, or gate their phospholipid permeation pathways remains unclear. Here we identify an inner activation gate, which is established by three hydrophobic residues, F518, Y563 and I612, in the middle of the phospholipid permeation pathway of TMEM16F-CaPLSase. Disrupting the inner gate profoundly alters TMEM16F phospholipid permeation. Lysine substitutions of F518 and Y563 even lead to constitutively active CaPLSases that bypass $Ca^{2+}$-dependent activation. Strikingly, an analogous lysine mutation to TMEM16F-F518 in TMEM16A (L543K) is sufficient to confer CaPLSase activity to the $Ca^{2+}$-activated $Cl^-$ channel (CaCC). The identification of an inner activation gate can help elucidate the gating and permeation mechanism of TMEM16 CaPLSases and channels.

[1] Department of Biochemistry, Duke University Medical Center, Durham, NC, USA. [2] Department of Chemistry, University of Massachusetts, Amherst, MA, USA. [3] Department of Biochemistry and Molecular Biology, University of Massachusetts, Amherst 01003 MA, USA. [4] Department of Neurobiology, Duke University Medical Center, Durham 27710 NC, USA. Correspondence and requests for materials should be addressed to H.Y. (email: huanghe.yang@duke.edu)

Cell surface exposure of phosphatidylserine (PS), a major phospholipid species that resides exclusively in the inner leaflet of the plasma membrane, serves as an important physiological signal that triggers various cellular processes, including phagocytosis, blood coagulation, bone mineralization, cell–cell fusion, and viral infection[1]. The CaPLSases from the TMEM16/ANO family are major phospholipid transporters that catalyze PS externalization in response to intracellular $Ca^{2+}$ elevation[1–8]. TMEM16F, one of the most well characterized CaPLSases, has been shown to be involved in blood coagulation[2,9], bone development[10], and HIV infection[11]. Loss-of-function mutations in TMEM16F cause Scott syndrome, a hemostatic disorder due to deficient PS externalization in platelets[2,12]. In addition, single-nucleotide polymorphisms (SNPs) of *TMEM16F/ANO6* are associated with the etiology of ankylosing spondylitis[13]. Besides TMEM16F, mutations in TMEM16E, another CaPLSase of the TMEM16 family, have also been implicated in a number of inherited diseases including gnathodiaphyseal dysplasia[14], limb-girdle muscular dystrophy[7,15–17], and Miyoshi myopathy[15–17]. Considering their importance in health and disease, a comprehensive understanding of the structure and function of the mammalian TMEM16 CaPLSases would facilitate drug discovery for these therapeutic targets.

An X-ray structure of a TMEM16 homolog from the fungus *Nectria haematococca* (nhTMEM16) captured in an activated $Ca^{2+}$-bound state illuminates a homodimeric assembly of TMEM16 with each monomer containing ten transmembrane (TM) helices[4] (Fig. 1a). Recent structural studies on TMEM16A-CaCC[18,19] and TMEM16F-CaPLase[20] further indicate that mammalian TMEM16 proteins also adopt a double-barreled dimeric architecture. $Ca^{2+}$ binding to a highly conserved $Ca^{2+}$-binding site in TM helices 6–8 (refs. [4,21,22]) triggers ion or phospholipid permeation through a hydrophilic pathway/groove comprising TMs 3–8 (refs. [4,20,23–25]). Interestingly, the hydrophilic groove in the $Ca^{2+}$-bound nhTMEM16 structure is exposed to the lipid environment owning to the physical separation of TM4 and TM6, both of which are located at the periphery of the protein. This intricate architecture thus suggests that phospholipid headgroups can enter and move along the permeation pathway while maintaining their hydrophobic acyl tails in the hydrocarbon core of the membrane[4,26–28], consistent with the widely accepted credit-card reader model for phospholipid permeation[29] (Fig. 1b). Nevertheless, the credit-card reader model does not implicate a gating control mechanism, which is required to regulate passive phospholipid permeation through TMEM16 CaPLSases in response to $Ca^{2+}$ binding.

Recent computational and functional studies guided by the $Ca^{2+}$-bound nhTMEM16 structure have proposed three critical sites for controlling phospholipid scrambling[26–28]: an extracellular $S_E$ site formed by charged residues from TM3 and TM6, a central constriction site in the middle of the hydrophilic groove and a cytosolic $S_C$ site composed of charged residues in TM4 (Fig. 1a). Although the exact roles of the $S_C$ site and the central constriction site in nhTMEM16 gating are unclear, molecular dynamic (MD) simulations and mutagenesis studies have suggested that the network of charged amino acids within the $S_E$ site (Fig. 1a) can serve as a putative extracellular gate to control phospholipid permeation[26,28]. In this model, $Ca^{2+}$ binding to nhTMEM16 triggers rearrangements of these charged amino acids at the $S_E$ gate allowing the hydrophilic pathway to accommodate and permeate phospholipids. While the nhTMEM16 structure has provided an invaluable structural template for computational and functional studies to unravel the molecular mechanism of phospholipid scrambling and regulation[4,26–28], lack of structural information of the mammalian TMEM16 scramblases has precluded a comprehensive

mechanistic understanding of phospholipid scrambling, and casted questions on how mammalian TMEM16 CaPLSases gate their phospholipid permeation pathways in a tightly controlled fashion.

Combining structure-guided mutagenesis, quantitative measurements of CaPLSase activity, and atomistic MD simulations, we identify F518, Y563, and I612 residues, which are located in the middle of the TMEM16F phospholipid permeation pathway, form an inner activation gate to control phospholipid translocation. Amino acid substitutions at these inner gate residues profoundly affect TMEM16F scrambling activity. Remarkably, we find that F518K and Y563K mutations lead to constitutively active CaPLSases independent of $Ca^{2+}$ activation. Moreover, TMEM16A-L543K, a corresponding mutation of TMEM16F-F518K, confers the CaCC the capability to constitutively permeate phospholipids. Our identification of an inner activation gate in controlling TMEM16F-CaPLSase phospholipid permeation thus provides an important framework for the mechanistic understanding of the gating mechanism of mammalian TMEM16 CaPLSases.

## Results

**Homology models of TMEM16F predict an inner activation gate**. In order to study TMEM16F-CaPLSase gating, a closed state structural model is necessary. Given the high sequence conservation between mammalian TMEM16F and TMEM16A, and low sequence conservation between mammalian TMEM16F and fungal nhTMEM16 (Supplementary Fig. 1a), we hypothesized that the $Ca^{2+}$-free structure of a murine TMEM16A (PDB: 5OYG)[23] would be a more suitable template for modeling the closed state of TMEM16F (Supplementary Fig. 1b). The $Ca^{2+}$-free TMEM16A-derived homology model of TMEM16F clearly stays in a closed conformation with its hydrophilic groove constricted, which results in high free energy barriers for both water and phosphate permeation during atomistic simulations (Fig. 1d).

To build an open state TMEM16F homology model, we first attempted to build a structural model (Fig. 1c) based on the $Ca^{2+}$-bound structure of TMEM16A (PDB: 5OYB)[23], which shows that $Ca^{2+}$ binding straightens TM6 around the glycine hinge to trigger TMEM16A activation (Supplementary Fig. 1b). Due to this $Ca^{2+}$-induced conformational change, the pore in the $Ca^{2+}$-bound TMEM16F model becomes more accessible to water permeation in comparison to the $Ca^{2+}$-free closed state (Fig. 1d, e). However, there remains a major constriction for phosphates near the center of the groove (Fig. 1d, f), suggesting that the 5OYB-derived TMEM16F model may instead represent an intermediate, partially open or inactivated state. Hence, we assign this 5OYB-derived TMEM16F model to an intermediate state to distinguish it from the $Ca^{2+}$-free closed state and the $Ca^{2+}$-bound fully open state.

To obtain a fully open TMEM16F structural model, we sought to utilize the $Ca^{2+}$-bound nhTMEM16 structure (PDB: 4WIS)[4], whose hydrophilic groove is widely opened (Fig. 1a and Supplementary Fig. 1c). However, owing to the low sequence homology between nhTMEM16 and TMEM16F, particularly outside of the TM region (Supplementary Fig. 1a), we started with the 5OYB-derived TMEM16F intermediate model and used steered MD to gradually move its TMs 3, 4, and 6 (Fig. 1c), whose conformations are most distinct between the TMEM16A and the nhTMEM16 structures[23], to mimic the positions observed in the open nhTMEM16 structure (Supplementary Movie 1) (see Methods for details). The final TMEM16F model thus harbors a widened hydrophilic groove, which we believe would represent an open conformation (Fig. 1c). Subsequent atomistic simulations of

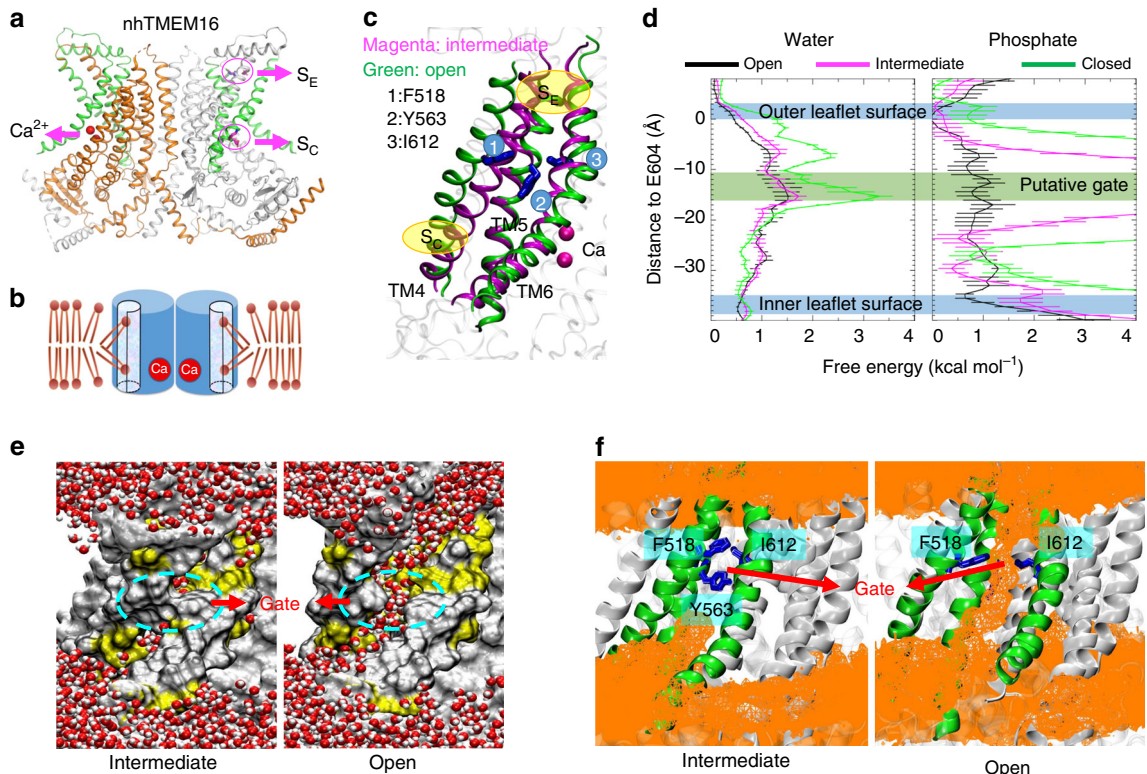

**Fig. 1** F518, Y563, and I612 form a putative inner activation gate for TMEM16F-CaPLSase. **a** X-ray structure of nhTMEM16 (PDB: 4WIS). The two monomers are colored in grey and brown, respectively. The transmembrane (TM) helices 4–6 lining the interior of the hydrophilic groove are highlighted in green. The bound $Ca^{2+}$ ions are represented as red spheres. Previously proposed $S_E$ and $S_C$ sites are marked with magenta circles. **b** A schematic representation of the credit-card reader model for phospholipid permeation through CaPLSases. **c** Superposition of TMs 4–6 in the predicted intermediate and open state models of TMEM16F. The intermediate was derived from the $Ca^{2+}$-bound TMEM16A structure (PDB: 5OYB), and the open configuration was derived from the hybridization of the $Ca^{2+}$-bound TMEM16A structure (PDB: 5OYB) and the $Ca^{2+}$-bound nhTMEM16 structure (PDB: 4WIS) (see Methods for details). Side chains of the putative inner activation gate residues are shown in blue sticks and numbered as 1, 2, and 3, respectively. **d** Effective free energy profiles of water (left) and phosphates (right) along the hydrophilic groove derived from the average densities from the last 100 ns of the 400 ns atomistic simulations of TMEM16F in open (black), intermediate (magenta), and closed (green) states. The y-axis is the relative distance from E604, a residue at the outer leaflet surface. The standard errors were estimated from the calculated averages over three independent simulations. **e** Snapshots from the atomistic simulations of the open or intermediate TMEM16F homology models. The protein is represented as silver surfaces and all polar residues in TMs 4–6 are colored in yellow. Waters within 15 Å of the protein are represented as stick-and-balls. **f** Phosphate densities near TMs 4–6 (green cartoons) derived from atomistic simulations. The phosphate accessible region (probability > 0.005) is represented as orange grid. Three putative gate residues (F518, Y563, I612) are represented as blue sticks

the open structural model verify that the final TMEM16F model is energetically stable (Supplementary Fig. 2a, b). More importantly, this hydrophilic groove becomes readily accessible to both water and especially phosphates (Fig. 1d–f), supporting the notion that this open state model could represent an active TMEM16F-CaPLSase. Interestingly, despite the limited simulation timeframe of 400 ns, we observed a total of three phosphatidylcholine externalization events (one in each run) via the widened hydrophilic groove, closely recapitulating the well-known credit-card reader transport mechanism[29] (Supplementary Fig. 2c and Movie 2).

After carefully scrutinizing the closed, intermediate and fully open models of TMEM16F and their capabilities to permeate water and phospholipids (Fig. 1d–f), we find that the constriction impeding the permeation of phospholipid headgroups mainly originates from three hydrophobic residues, F518, Y563, and I612, whose side chains point toward the central axis of the hydrophilic groove (Fig. 1d, f). Indeed, in the closed and intermediate states, the side chains of these residues stay in close proximity to create a physical barrier restricting phospholipid permeation, whereas in the $Ca^{2+}$-bound fully open state, they separate to widen the hydrophilic groove, thus permitting

phospholipid permeation (Fig. 1d–f and Supplementary Fig. 2c). Based on these observations, we hypothesize that F518, Y563 and I612 constitute an inner activation gate of the mammalian TMEM16F-CaPLSase.

**An imaging assay to quantify CaPLSase activity.** A microscopy assay utilizing fluorescently conjugated Annexin V (AnV), a PS-binding protein, to detect CaPLSase-mediated PS surface exposure enables quantification of phospholipid scrambling at a single-cell resolution[30]. In a recent study, we optimized this assay by genetically ablating endogenous TMEM16F expression, which eliminates endogenous CaPLSase interference in HEK293T cells[31]. We used the TMEM16F-null cell line throughout this study to quantify the scrambling activities of the heterologously expressed wild type (WT) and mutant murine TMEM16F (mTMEM16F) CaPLSases in real time. Heterologous expression of WT mTMEM16F into the TMEM16F-null HEK293T cells via transient transfection induced strong CaPLSase activity with minimal sign of apoptosis[31] (Fig. 2a, $t = 0$ min and Supplementary Fig. 3a). We limited our CaPLSase measurements to the first 10 min after ionomycin stimulation, during

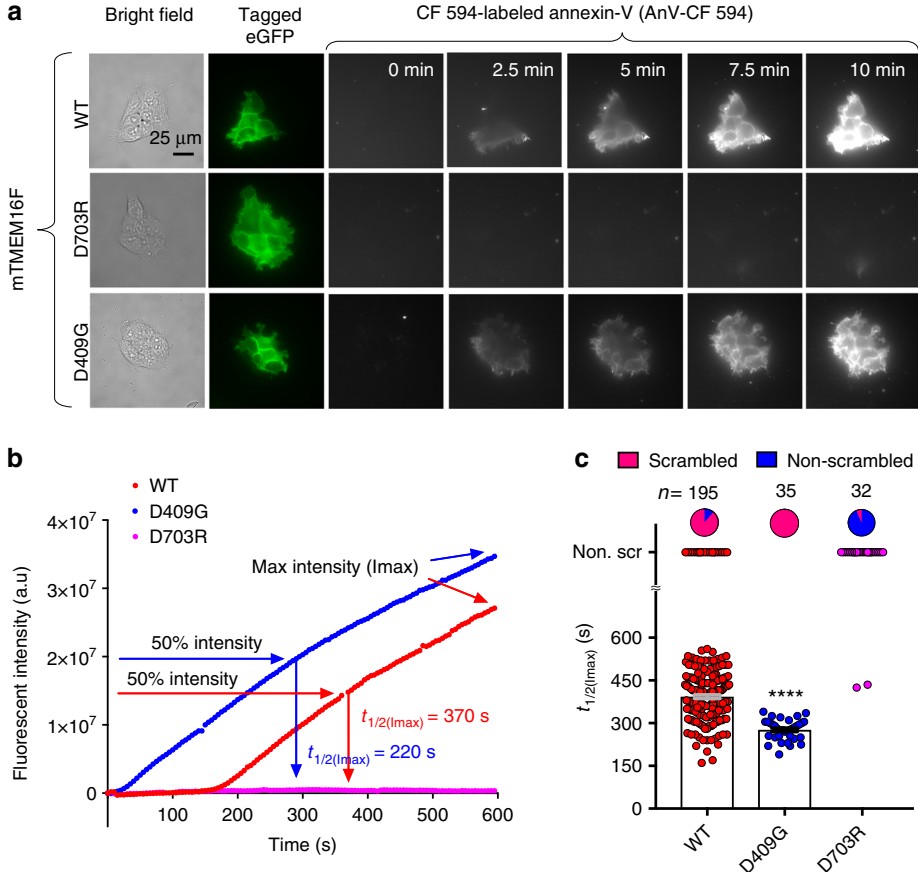

**Fig. 2** Quantification of CaPLSase activity using an Annexin V (AnV)-based imaging assay. **a** Representative images of ionomycin-induced scrambling activity of TMEM16F-knockout HEK293T cells expressing eGFP-tagged TMEM16F-WT, D409G, or D703R. CF 594-tagged-AnV signal representing phospholipid scrambling was recorded by time-lapsed imaging for 10-min post 5 μM ionomycin application (0 min) at a 5-s acquisition interval. Scale bars, 25 μm. **b** Representative AnV fluorescent intensity changes for TMEM16F-WT, D703R, and D409G expressing cells in **a**. $t_{1/2(Imax)}$ is the time at which each cell reaches 50% of its maximum AnV fluorescent intensity. **c** Ionomycin-induced CaPLSase activity for TMEM16F-WT, D409G, and D703R as quantified by $t_{1/2(Imax)}$. Each data point represents one cell and $n$ is the total number of cells analyzed. Cells that did not exhibit ionomycin-induced CaPLSase activity were denoted as non-scr (non-scrambling) and are excluded from statistical analysis. As a majority of D703R expressing cells does not scramble, mean and SEM are not assigned for this mutation. The pie charts illustrate the percentage of scrambling cells in response to ionomycin stimulation. Statistical analysis was performed using unpaired two-sided Student's $t$-test for WT and D409G. ****: $p < 0.0001$. Error bars indicate SEM. All of the experiments based on AnV labeling were done in $Ca^{2+}$-containing buffer. Source data are provided in the Source Data file

which the interference from apoptosis-induced phospholipid scrambling is negligible[31]. Moreover, stable expression of mTMEM16F and treatment with pan-caspase inhibitor Q-VD-OPh show no difference in scrambling activity compared to that of transiently expressed TMEM16F cells (Supplementary Fig. 3b), further suggesting that transient transfection and overexpression of TMEM16F does not induce apoptosis-dependent phospholipid scrambling. To circumvent the variations in scrambling activity due to the differences in TMEM16F expression levels and ionophore-mobilized intracellular $Ca^{2+}$ levels, we use $t_{1/2(Imax)}$, the time at which each cell reaches 50% of its maximum AnV fluorescent intensity, instead of the maximum AnV fluorescent intensity to quantify CaPLSase activity[31] (Fig. 2b, c). Using this assay, we confirmed that D409G, a TMEM16F gain-of-function mutation[2], displayed an accelerated phospholipid scrambling as seen by its significantly shorter $t_{1/2(Imax)}$ in comparison to that of WT mTMEM16F (Fig. 2b, c). Without ionomycin treatment, D409G expression did not induce spontaneous PS exposure (Fig. 2a, 0 min and Supplementary Fig. 3c), suggesting that this mutant CaPLSase is not constitutively activated in HEK293T cells. On the other hand, the $Ca^{2+}$-binding site mutation D703R, whose equivalent mutation in TMEM16A

completely eliminated $Ca^{2+}$-dependent activation of the CaCC[21], abolished phospholipid scrambling activity ($t_{1/2(Imax)}$ value assigned as infinity and denoted as non-scrambling) (Fig. 2a–c). Taken together, our optimized CaPLSase imaging assay provides us a faithful method to characterize the putative inner gate residues of TMEM16F.

**Ala mutations at the inner gate promote TMEM16F-CaPLSase.** To examine the roles of the putative gate residues in regulating phospholipid movement across the hydrophilic groove, we first replaced these bulky residues with Ala. We hypothesized that the smaller and less hydrophobic sidechain from Ala would widen the inner gate to alleviate the constrictive effect, thereby facilitating phospholipid permeation. Using the optimized CaPLSase assay, we found that all three Ala mutations of F518, Y563, and I612 exhibited strong gain-of-function phenotypes (Fig. 3a, b). When quantified with $t_{1/2(Imax)}$, F518A and I612A showed significantly enhanced CaPLSase activity compared to WT-TMEM16F. Consistent with our observation, a recent report also shows an enhanced TMEM16F scrambling activity by the I612A mutation[32].

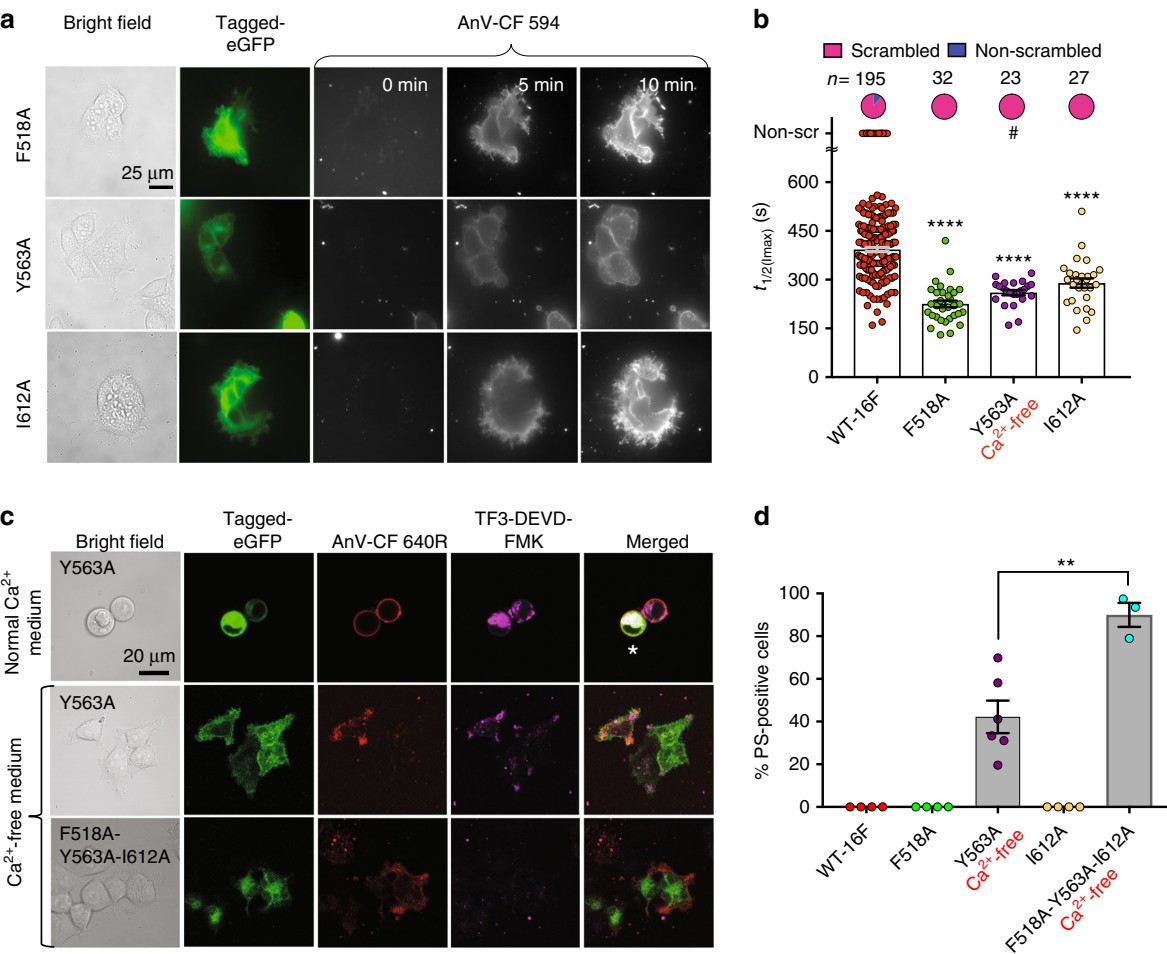

**Fig. 3** Alanine mutations of the inner gate residues promote TMEM16F activation. **a** Representative images of ionomycin-induced scrambling activity of TMEM16F-knockout HEK293T cells expressing eGFP-tagged TMEM16F-F518A, Y563A, or I612A. Y563A was expressed in $Ca^{2+}$-free medium to minimize cytotoxicity, and its CaPLSase activity was measured in $Ca^{2+}$-containing buffer. AnV-CF 594 signal representing phospholipid scrambling was recorded by time-lapsed imaging for 10-min post 5 μM ionomycin application (0 min) at a 5-s acquisition interval. Scale bars, 25 μm. **b** Ionomycin-induced CaPLSase activity for wild type (WT) and the mutant TMEM16F quantified by $t_{1/2(Imax)}$. Each data point represents one cell and n denotes the total number of cells measured. The pie charts illustrate the percentage of ionomycin-induced scrambling cells. The cells that did not exhibit ionomycin-induced CaPLSase activity were denoted as non-scr (non-scrambling) and were excluded from statistical analysis. Statistical analysis was performed using one-way ANOVA with Tukey's multiple comparisons test. ****: $p < 0.0001$. Error bars indicate SEM. # denotes mutations with spontaneous CaPLSase activity. **c** Representative images of eGFP-tagged Y563A- and F518A-Y563A-I612A-expression in TMEM16F-knockout HEK293T cells without ionomycin stimulation. CF 640R-tagged-AnV (AnV-CF 640 R) labels PS-positive cells. White asterisk labels apoptotic cells with positive AnV, strong cytosolic TF3-DEVD-FMK (indicative of cleaved caspases 3/7) staining, and rounded morphology. Scale bars, 20 μm. **d** Percentage of cells with spontaneous CaPLSase activities without ionomycin stimulation. Each data point represents the percentage of PS-positive cells from the total TMEM16F-expressing cells in one coverslip (see Methods for details). Cells expressing Y563A and F518A-Y563A-I612A were cultured in $Ca^{2+}$-free medium to avoid cytotoxicity. Statistical analysis was performed using unpaired two-sided Student's t-test. **: $p = 0.0047$ (<0.01). Error bars indicate SEM. All of the experiments based on AnV labeling were done in $Ca^{2+}$-containing buffer. Source data are provided in the Source Data file

Strikingly, we found that Y563A expression in regular $Ca^{2+}$-containing medium resulted in constitutively exposed PS even in the absence of ionomycin stimulation (Fig. 3c). Rounded cell morphology and strong caspase 3/7 activity staining (TF3-DEVD-FMK signal) further suggest that overexpressing Y563A likely induces cytotoxicity (Fig. 3c). Interestingly, a rounded cell morphology has also been observed when a gain-of-function mutation of TMEM16E is heterologously expressed in HEK293 cells, suggesting that gain-of-function CaPLSases tend to induce cytotoxicity[33]. On the other hand, overexpression of TMEM16F D409G gain-of-function mutation[2] in the TMEM16F deficient HEK293T cells only resulted in accelerated ionomycin-induced phospholipid scrambling (Fig. 2) without apparent changes in cell morphology and constitutive PS exposure in regular cell culture medium (Supplementary Fig. 3c). This result is different from the observation of constitutive PS exposure when D409G was expressed in lymphoma cells[34]. We speculate that this discrepancy may be due to the differences in basal $Ca^{2+}$ activities and cellular regulations in the two different cell types. Thus, Y563A-induced cytotoxicity in $Ca^{2+}$ containing medium suggests that Y563A might be a more potent gain-of-function mutation compared with D409G.

To test whether this cytotoxicity was due to the excessive $Ca^{2+}$-dependent activation of TMEM16F, we cultured Y563A-expressing cells in a $Ca^{2+}$-free medium to suppress the basal $Ca^{2+}$ activity. In this condition, we observed about 42% of the Y563A-expressing cells still exhibited spontaneous PS surface exposure (Fig. 3c, d). Notably, these PS-positive cells possessed healthy, polarized morphology and displayed minimal caspase activity, all of which indicate that they were viable and that

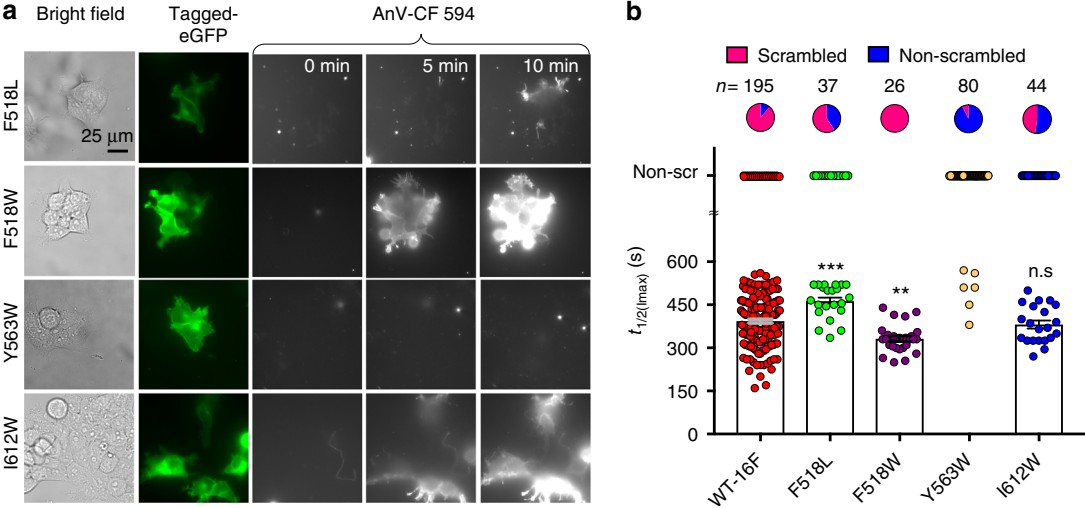

**Fig. 4** Bulky hydrophobic mutations at the inner gate hinder TMEM16F lipid permeation. **a** Representative images of ionomycin-induced scrambling activity of TMEM16F-knockout HEK293T cells expressing eGFP-tagged TMEM16F-F518L, F518W, Y563W, or I612W. CF 594-tagged-AnV signal representing phospholipid scrambling was recorded by time-lapsed imaging for 10-min post 5 μM ionomycin application (0 min) at a 5-s acquisition interval. Scale bars, 25 μm. **b** Ionomycin-induced CaPLSase activity for WT- and mutant- TMEM16F quantified by $t_{1/2(Imax)}$. Each data point represents one cell and $n$ is the total number of cells measured. The pie charts illustrated the percentages of ionomycin-induced scrambling cells. The cells that did not exhibit ionomycin-induced CaPLSase activity were denoted as non-scr (non-scrambling) and are excluded from statistical analysis. As a majority of Y563W-expressing cells does not scramble, mean and SEM are not assigned for this mutation. Statistical analysis was performed using one-way ANOVA with Tukey's multiple comparisons test. ***: $p = 0.0006$; **: $p = 0.0013$; ns: not significant ($p = 0.9167$). Error bars indicate SEM. All of the experiments based on AnV labeling were done in $Ca^{2+}$-containing buffer. Source data are provided in the Source Data file

TMEM16F-Y563A expression promotes spontaneous PS externalization. While the remaining (58%) Y563A-expressing cells did not show spontaneous CaPLSase activity, likely due to lower expression levels, they did exhibit gain-of-function phenotype upon ionomycin stimulation as evidenced by a smaller $t_{1/2(Imax)}$ value in comparison to that of TMEM16F-WT (Fig. 3a, b). These observations suggest that, distinct from D409G, which is not spontaneously active (Fig. 2a), Y563A is a potent gain-of-function mutation that can facilitate strong CaPLSase activity under resting intracellular $Ca^{2+}$. Consistent with its increased CaPLSase activity, we found that Y563A also significantly enhanced its ion channel activation by $Ca^{2+}$ (Supplementary Fig. 4a–d). In addition, Y563A completely abolishes TMEM16F channel desensitization or rundown in addition to altering ion selectivity (Supplementary Fig. 4e, f). All taken together, these results further support that Y563 plays a critical role in gating TMEM16F substrate permeation.

As F518A, Y563A, and I612A all exhibited pronounced gain-of-function phenotypes in their scrambling activity, we test whether their effects on phospholipid permeation can be additive. We therefore generated a triple Ala mutation, F518A-Y563A-I612A, and interrogated its $Ca^{2+}$-dependent phospholipid scrambling. Interestingly, we found that this triple mutation displayed more enhanced gain-of-function phenotype than the single Ala mutations (Fig. 3c, d). Specifically, more than 90% of the triple mutation-expressing cells exhibited constitutive CaPLSase activity in the absence of ionomycin treatment. The observed additive effect of the triple Ala mutation suggests a possibility that the three putative inner gate residues work synergistically to control phospholipid movement across the hydrophilic groove (Fig. 1d, f).

**Bulky mutations at the inner gate hinder TMEM16F-CaPLSase.** Having established the critical roles of the putative inner gate residues in phospholipid gating, we next tested whether

replacing these residues with a bulkier amino acid could disrupt the physical constriction and affect CaPLSase activity. A recent study identifies residues that are important for lipid scrambling in nhTMEM16 via Trp substitutions[28]. Taking a similar approach, we found that Trp substitutions of Y563 and I612, Y563W and I612W, markedly impaired phospholipid scrambling of TMEM16F, giving rise to loss-of-function scramblases (Fig. 4a, b). Almost all Y563W-expressing cells and ~52% of I612W-expressing cells showed no ionomycin-induced CaPLSase activity. Paradoxically, F518W slightly enhanced TMEM16F phospholipid permeation (Fig. 4a, b). We suggest that this enhancement in scrambling of F518W is likely due to the higher hydrophilicity of Trp compared with that of Phe (the hydropathy indexes for Trp and Phe are −0.9 and 2.8, respectively[35]). This mild gain-of-function phenotype of F518W is further corroborated by a similar observation that an equivalent mutation in nhTMEM16 also enhanced phospholipid scrambling[28]. To examine the significance of the hydrophobicity at F518 for phospholipid permeation, we substituted F518 with Leu, a more hydrophobic residue with the hydropathy index of 3.8 (ref. [35]). In contrast to F518W, F518L exhibited a loss-of-function phenotype with a reduced scrambling activity (Fig. 4a, b). Furthermore, we found that a large population of F518L-expressing cells showed no CaPLSase activity whereas cells that did scramble exhibited significantly prolonged $t_{1/2(Imax)}$ values compared with that of WT-TMEM16F (Fig. 4b). Taken together, our findings highlight the importance of both size and hydrophobicity at the inner gate, both of which are critical in controlling phospholipid permeation.

**Hydrophilic and charged mutations at the inner gate.** To further dissect the role of the inner gate in controlling phospholipid permeation, we disrupted the hydrophobicity at the inner gate by substituting the inner gate residues with either hydrophilic or charged amino acids. Surprisingly, all of these mutations turned out to be gain-of-function (Fig. 5 and Supplementary Fig. 5).

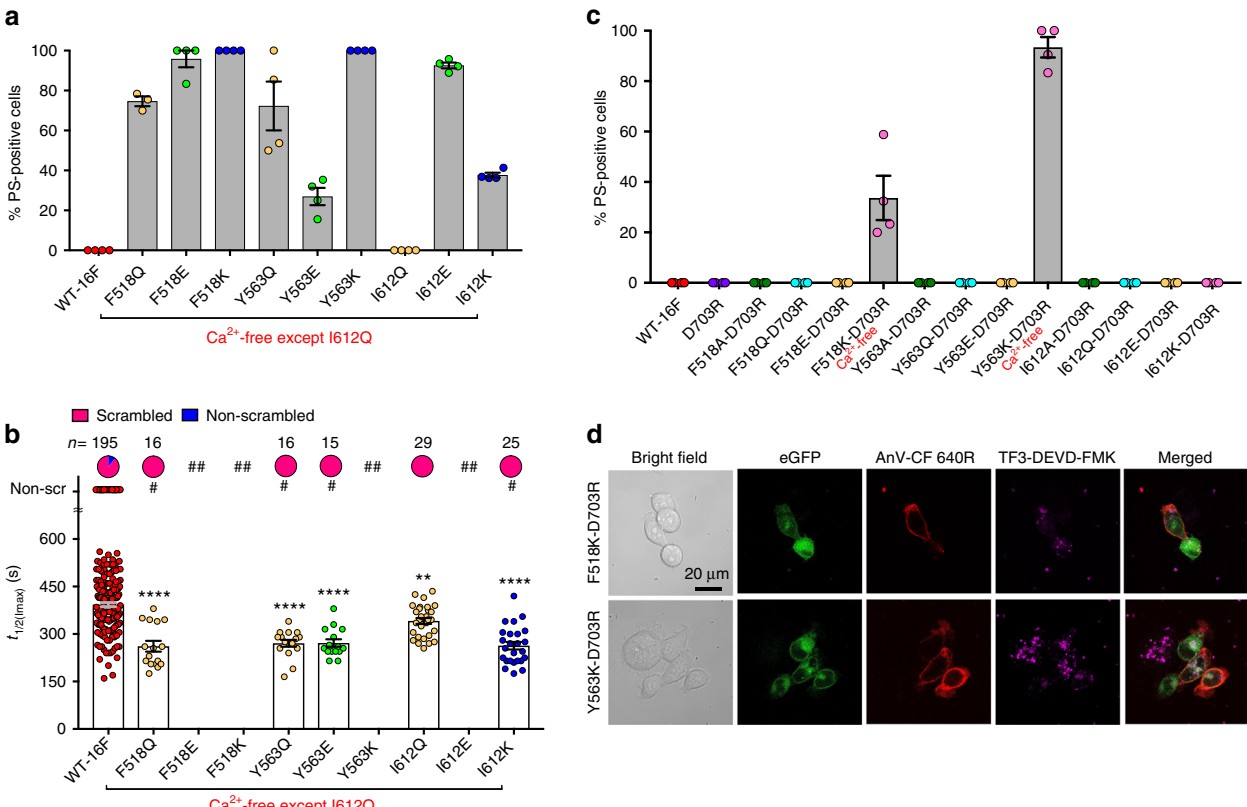

**Fig. 5** Hydrophilic or charged mutations at the inner gate make TMEM16F constitutively open. **a** Except I612Q, all inner gate mutations with polar and charged side chains were constitutively active with spontaneous CaPLSase activities. Each dot represents the percentage of PS-positive cells over the total expressing cells (either WT-TMEM16F or mutants) in one coverslip (see Methods for details). Except I612Q, all other TMEM16F mutants were expressed in $Ca^{2+}$-free medium to suppress their excessive scrambling activity and cytotoxicity. **b** Ionomycin-induced CaPLSase activity for wild type (WT) and the mutant TMEM16F quantified by $t_{1/2(Imax)}$. The constitutively activated TMEM16F mutants that caused >80% of spontaneous PS exposure (##) were excluded from this analysis. For the modest gain-of-function mutations (<80% spontaneous PS exposure, #), only the cells that did not show spontaneous PS exposure were analyzed for their $t_{1/2(Imax)}$ values after ionomycin stimulation. Each dot represents one cell, and n is the total number of cells measured. The pie charts illustrate the percentages of ionomycin-induced scrambling cells, and non-scr denotes non-scrambling. Statistical analysis was performed using one-way ANOVA with Tukey's multiple comparisons test. **: $p = 0.0093$; ****: $p < 0.0001$. Error bars indicate SEM. **c** The impaired $Ca^{2+}$ binding site mutation D703R suppressed all constitutively activated TMEM16F CaPLSases except F518K and Y563K. Each dot represents the percentage of PS-positive cells over the total expressing cells in one coverslip. **d** Representative images of eGFP-tagged F518K-D703R- and Y563K-D703R-expressing in TMEM16F-knockout HEK293T cells in the absence of ionomycin stimulation. AnV-CF 640 R labeled PS-positive cells. Weak TF3-DEVD-FMK staining of cleaved caspases 3/7 suggests that the cells are not apoptotic. Both F518K-D703R and Y563K-D703R (in **c** and **d**) were expressed in $Ca^{2+}$-free medium to avoid cell death and cytotoxicity, while all other D703R-coupled mutations (in **c**) were expressed in normal $Ca^{2+}$ medium. All experiments were done in the TMEM16F-knockout HEK293T cells. All of the experiments based on AnV labeling were done in $Ca^{2+}$-containing buffer. Scale bar, 20 μm. Source data are provided in the Source Data file

Furthermore, with the exception of I612Q, which showed the modest gain-of-function phenotype, all other tested mutations exhibited constitutive scrambling activities (Fig. 5a and Supplementary Fig. 5a). Similar to Y563A and the triple Ala mutation (Fig. 3), all of the spontaneous scrambling mutations from the hydrophilic/charged substitutions, except I612Q, induced cell stress and cytotoxicity when cultured in regular $Ca^{2+}$ medium. To verify that the cytotoxicity was due to $Ca^{2+}$-dependent activation of TMEM16F, we cultured the transfected cells in the $Ca^{2+}$-free medium. Removal of extracellular $Ca^{2+}$ alleviated the cytotoxicity from most of the spontaneous gain-of-function mutations, except for the two most potent ones, F518K and Y563K (Fig. 5a and Supplementary Fig. 5a). In fact, these two mutations were highly active that all of the expressing cells were PS-positive with apparent signs of apoptosis (rounded morphology and strong caspase 3/7 activities) even when cultured in $Ca^{2+}$-free medium (Supplementary Fig. 5a). Cells expressing other spontaneous gain-of-function mutations were healthy in

the $Ca^{2+}$-free medium and exhibited various levels of spontaneous and ionomycin-induced CaPLSase activities (Fig. 5a, b and Supplementary Fig. 5b). Interestingly, negatively charged substitution to the inner gate residues, especially F518E and I612E, also caused a large population of transfected cells to display spontaneous PS externalization (Fig. 5a and Supplementary Fig. 5a). Thus, these results suggest that increasing hydrophilicity or introducing charges at the inner activation gate promotes phospholipid permeation.

Besides lowering the permeation energy barrier by facilitating phospholipid headgroups interaction, the hydrophilic and charged residues introduced at the inner gate may also shift the equilibrium between the open and closed states of TMEM16F-CaPLSase by destabilizing the closed state and/or stabilizing the open state. Other than directly affecting the inner gate itself, the mutations at the putative inner gate may allosterically enhance $Ca^{2+}$-binding to give rise to the gain-of-function phenotype. To examine this possibility, we couple our polar and charged inner

gate mutations with the $Ca^{2+}$-binding site mutation D703R (Fig. 2). Interestingly, abrogating $Ca^{2+}$ binding successfully suppresses the gain-of-function phenotypes for most of the polar and charged inner gate mutations (except F518K and Y653K) even in the regular $Ca^{2+}$-containing medium (Fig. 5c and Supplementary Fig. 6). Since all of the double mutations with D703R have suffice cell surface expression (Supplementary Fig. 6), membrane targeting is unlikely to be the reason why these double mutations lost their spontaneous CaPLSase activities. We speculate that the resting, basal $Ca^{2+}$ level (below 100 nM) is sufficient to trigger the opening of the activation gate in the corresponding single gain-of-function mutations to induce spontaneous CaPLSase activity (Fig. 4a).

Notably, we found that disrupting $Ca^{2+}$ binding by the D703R mutation did not prevent spontaneous scrambling of F518K and Y563K (Fig. 5c, d). In fact, about 34% of F518K-D703R and nearly 93% Y563K-D703R-expressing cells still exhibit spontaneous PS exposure in $Ca^{2+}$-free medium. The constitutively active F518K-D703R and Y563K-D703R mutations further support that the positive charges at these two inner gate residues destabilize the closed state of the phospholipid pathway independent of $Ca^{2+}$ binding. I612K in TM6, however, when coupled with D703R, failed to spontaneously expose PS (Fig. 5c), suggesting that the charge on this residue imposes a weaker effect in destabilizing the activation gate than those of F518K and Y563K in TM4 and TM5, respectively. The results also illustrate that positive charges introduced at these two positions allow TMEM16F phospholipid pathway to stay constitutively open

without $Ca^{2+}$ binding. Although we could not investigate the $Ca^{2+}$ binding affinity of TMEM16F mutations to completely rule out the possibility that these gain-of-function mutations allosterically enhance $Ca^{2+}$ binding, the constitutive scrambling observed in F518K-D703R and Y563K-D703R mutations, whose $Ca^{2+}$ binding is severely impaired, strongly indicates that these charged residues introduced at the inner activation gate directly open the phospholipid pathway without much facilitation from $Ca^{2+}$ binding.

**An equivalent inner gate controls TMEM16A gating.** As TMEM16F-CaPLSase and TMEM16A-CaCC share a high degree of sequence similarity in their TMs (Supplementary Fig. 1a), we tested whether equivalent residues in TMEM16A could also serve as an inner activation gate. Interestingly, unlike WT-TMEM16A[21], our inside-out patch clamp recordings show that both TMEM16A-L543K and I637K mutant channels could be activated by membrane depolarization in the absence of intracellular $Ca^{2+}$ (Fig. 6a, b), suggesting that introducing positive charges to these two hydrophobic residues in TM4 and TM6 could unlatch the gate and promote TMEM16A channel opening. Among all three mutations, S588K showed the weakest effect on TMEM16A voltage-dependent activation despite being activated by $Ca^{2+}$ (Supplementary Fig. 7a).

More strikingly, similar to TMEM16F-F518K (Fig. 5a), the corresponding mutation TMEM16A-L543K induced cytotoxicity when the transfected cells were cultured in regular

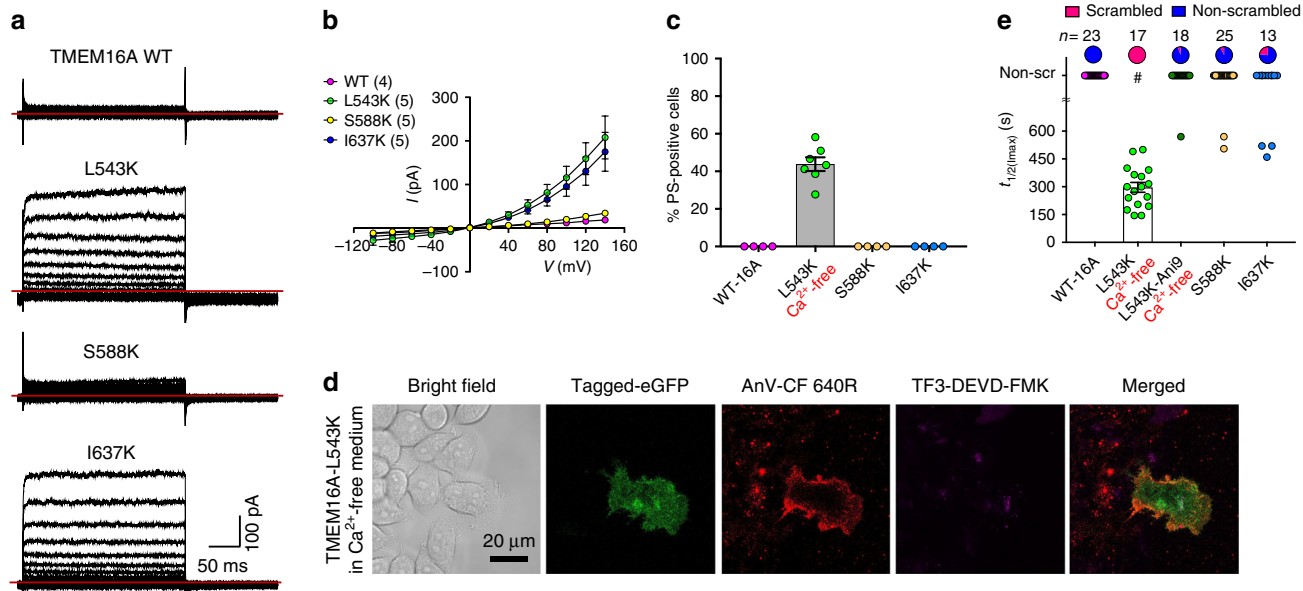

**Fig. 6** A Lysine mutation at TMEM16A inner gate converts the CaCC into a CaPLSase. **a** Representative current traces of inside-out patches excised from TMEM16A-WT or mutations-expressing HEK293T cells in the absence of intracellular $Ca^{2+}$. Testing potentials were from −100 mV to +140 mV with 20 mV increments. Both holding and repolarizing potentials were −60 mV. **b** Current–voltage (I–V) relationship of the equivalent inner gate mutations in TMEM16A. Error bars indicate SEM. **c** L543K-TMEM16A expression induced spontaneous PS exposure (in the absence of ionomycin treatment) even when cultured in $Ca^{2+}$-free medium (#). Each dot represents the percentage of PS-positive cells over the total expressing cells (WT-TMEM16A or mutants) of one coverslip. **d** Representative images of eGFP-tagged L543K-TMEM16A-expressing cells without ionomycin stimulation. AnV-CF 640R-labeled PS-positive cells. Weak TF3-DEVD-FMK staining of cleaved caspases 3/7 suggests that the cells were not apoptotic. **e** Ionomycin-induced CaPLSase activity for wild type (WT) and the mutant TMEM16A quantified by $t_{1/2(Imax)}$. Treatment with 10 μM Ani9 inhibits TMEM16A-L54K scrambling activity as indicated by its $t_{1/2(Imax)}$. Each dot represents one cell and $n$ is the total number of cells measured. The pie charts illustrated the percentages of ionomycin-induced scrambling cells and non-scr denotes non-scrambling and are excluded from statistical analysis. As a majority of TMEM16A-WT, -L543K (Ani9), S588K, and I637K expressing cells do not scramble, mean and SEM are not assigned for those mutations. Error bars indicate SEM. All imaging experiments were done in the TMEM16F-knockout HEK293T cells. All of the experiments based on AnV labeling were done in $Ca^{2+}$-containing buffer. Scale bars, 25 μm. Source data are provided in the Source Data file

$Ca^{2+}$-containing medium (Supplementary Fig. 7b). However, culturing these cells in a $Ca^{2+}$-free medium alleviated cytotoxicity; yet about 44% of the L543K-TMEM16A-expressing cells still exhibited constitutive PS exposure as indicated by the positive AnV signal (Fig. 6c, d). Their morphology as well as activated caspase 3/7 level were similar to the healthy non-transfected cells (Fig. 6d). For the L543K-expressing cells that did not show spontaneous PS exposure, application of a lower concentration of ionomycin (2.5 μM) rapidly induced PS exposure (Fig. 5e and Supplementary Fig. 7c), consistent with TMEM16A having a higher $Ca^{2+}$ sensitivity than TMEM16F[9]. These results clearly demonstrate that L543K, a single point mutation at the inner activation gate, is capable of converting TMEM16A-CaCC into a CaPLSase. In addition, a potent TMEM16A-CaCC blocker Ani9 (ref. [36]) effectively blocked ionomycin-induced phospholipid scrambling through L543K (Fig. 6c), confirming that this mutation rendered TMEM16A the capability to permeate phospholipids. Interestingly, TMEM16A-L543K also has altered ion selectivity (Supplementary Fig. 7b–d), further supporting the critical role of L543 residue in controlling ion permeation.

In contrast to L543K, neither S588K nor I637K expression caused spontaneous or ionomycin-induced PS exposure through TMEM16A (Fig. 6c, e and Supplementary Fig. 7c). As I637K shows a gain-of-function in its channel activation (Fig. 5a, b), the lack of CaPLSase activity indicates that while the positive charge at I637 can enhance ion permeation, its activation gate might not be opened widely enough to permeate phospholipids. This is also consistent with the milder gain-of-function phenotype observed in the equivalent I612K-TMEM16F-CaPLSase (Fig. 5a, c). In summary, our observation that the conversion of TMEM16A-CaCC to a constitutively activated CaPLSase by L543K further strengthens the importance of positive charges at the inner activation gate in promoting gating of TMEM16 proteins.

## Discussion

In this study, we identified an inner activation gate formed by three hydrophobic residues, F518, Y563, and I612, near the middle of the phospholipid permeation pathway of TMEM16F-CaPLSase. Side chain properties of the inner gate residues have profound effects on TMEM16F-CaPLSase activities. In particular, introducing positive charges to F518 and Y563 constitutively opens the activation gate to allow spontaneous phospholipid

permeation even in the absence of $Ca^{2+}$ binding. Moreover, the positive charge at the putative inner activation gate not only promotes phospholipid scrambling in TMEM16F but also enhances channel activation in TMEM16A, consistent with a possibility that they share a similar mechanism of activation. More strikingly, we found that a single point mutation, L543K, at the inner activation gate can convert the non-scrambling CaCC TMEM16A into a constitutively activated CaPLSase. This observation stands in contrast to a previous study, which shows that substitution of a long stretch of 34 amino acids at TMs 4–5 from TMEM16F is required to transform CaCC TMEM16A into a scramblase[30].

Based on our functional and computational studies, here we propose a clam shell model to describe the gating mechanism of TMEM16F-CaPLSase (Fig. 7). We speculate that the interface between TM4 and TM6 can open and close like a clam shell to control the accessibility of phospholipids to the interior of the hydrophilic groove, where phospholipid permeation is catalyzed (Supplementary Fig. 2c and Movie 2). F518 in TM4 and I612 in TM6 likely serve as gate-keepers for the opening of this interface. In the absence of $Ca^{2+}$, they stay close to each other to block the accessibility of phospholipid headgroups to the hydrophilic groove. Upon $Ca^{2+}$ binding, large-scale conformational changes, likely including the movement of TM6 (Supplementary Fig. 1b) around a conserved glycine hinge[23,37] and the rearrangement of TMs 3–5, separate TM4 and TM6 (Fig. 1c). These conformational rearrangements likely culminate in the opening of the TM4–TM6 interface as well as the inner gate residues. Subsequently, the interior of the hydrophilic groove becomes exposed to the surrounding phospholipids such that their phospholipid headgroups can enter and translocate via the credit-card reader mechanism. By projecting its bulky sidechain into the center of the hydrophilic groove, Y563 in TM5 likely serves as a cap that stabilizes the inner gate and further obstructs phospholipid permeation in the closed state. Dilation of the inner gate following TM4 and TM6 separation allows the Y563 hydroxyl group to interact with phospholipid headgroups and facilitate phospholipid permeation. Introducing smaller, polar or charged residues to these critical locations tends to remove steric hindrance, likely via destabilizing the TM4–TM6 interface to promote spontaneous opening transitions, resulting in enhanced phospholipid permeation. In the case of F518K and Y563K, their inner activation gates become so severely disrupted that phospholipids can freely

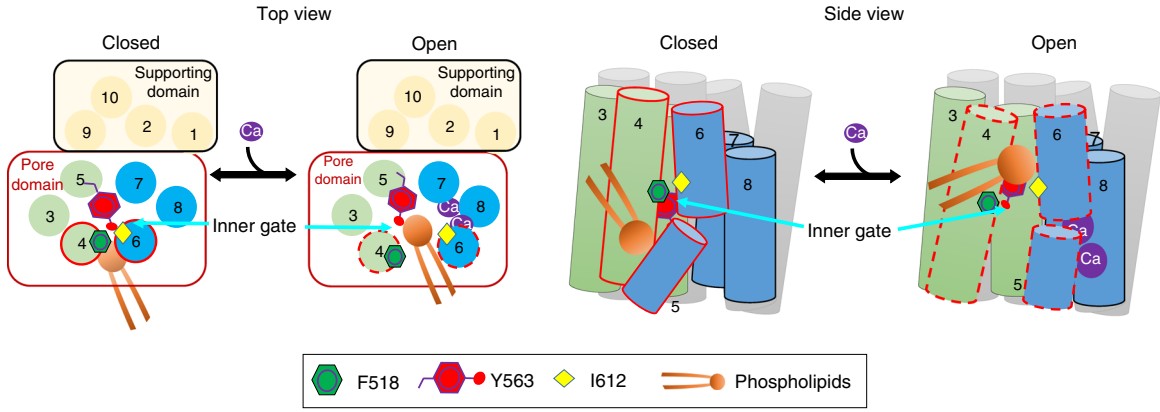

**Fig. 7** A clam shell model for TMEM16F-CaPLSase gating. In each subunit, TMs 3–8 enclose a hydrophilic phospholipid permeation pathway. The interface between TM4 and TM6 guards the inner activation gate, precluding phospholipid access to the hydrophilic pathway when $Ca^{2+}$ is absent. $Ca^{2+}$ binding triggers the separation of TM4 and TM6, leading to the opening of the inner activation gate and subsequent phospholipid permeation. Reducing size or increasing hydrophilicity of the inner activation gate residues weakens the TM4–TM6 interface, thereby enhances phospholipid permeation

go through the constitutively open gate in the absence of $Ca^{2+}$ binding.

While this work is under review, four independent structural characterizations of TMEM16 CaPLSases were released, reporting the structures of the fungal nhTMEM16, afTMEM16, murine TMEM16F (mTMEM16F), and human TMEM16K (hTMEM16K)[20,24,25,38]. Interestingly, the $Ca^{2+}$-free mTMEM16F structure exhibits high similarities to the murine TMEM16A structures[18,23], especially in the TM region including TM4 and TM6. This is consistent with our hypothesis that mammalian TMEM16F and TMEM16A share more structural similarities than fungal nhTMEM16 (Supplementary Fig. 1a). While the structures of mTMEM16F provide valuable insights into the mammalian CaPLSases, minimal conformational changes were observed between the $Ca^{2+}$-bound and the $Ca^{2+}$-free mTMEM16F or between mTMEM16F and the non-scrambling mTMEM16A. It thus remains unclear how TMEM16F-CaPLSase regulates phospholipid permeation based on these structural studies. Moreover, the extracellular segments of TM4 and TM6 in the $Ca^{2+}$-bound mTMEM16F structure are packed against each other preventing phospholipids from accessing the interior of the hydrophilic groove[20], suggesting that this structure may represent an intermediate conformation rather than a fully open and active conformation (Fig. 1e, f). On the other hand, widening of TM4-TM6 interface was observed in the $Ca^{2+}$-bound structures of fungal afTMEM16 (ref. [24]), nhTMEM16 (refs. [4,25]), and hTMEM16K[38], which is consistent with our proposed clam shell gating model in TMEM16F.

Identification of the inner activation gate for TMEM16F-CaPLSase also offers insights into our understanding of the gating mechanism of TMEM16-CaCCs. We suggest that, instead of utilizing the $Ca^{2+}$-dependent clam shell activation mechanism for TMEM16 CaPLSases, $Ca^{2+}$ binding to TMEM16-CaCCs may induce coordinated movement of TM4 and TM6 to dilate the ion permeation pathway, thereby allowing ions to go through while still maintaining a tight barrier at the TM4–6 interface, which excludes phospholipid permeation. This gating mechanism is supported by our observation that I637K-TMEM16A, which is a gain-of-function CaCC, does not facilitate phospholipid permeation (Fig. 6a–d). On the other hand, when a positive charge is introduced to L543 in TM4, the interaction between TM4 and TM6 might be dramatically weakened so that they may open like a clam shell to allow spontaneous permeation of phospholipids. We therefore postulate that TMEM16 CaPLSases and CaCCs may share evolutionarily conserved $Ca^{2+}$-dependent gating mechanism and overall similar design in their inner activation gates. Their distinct substrate selectivity and permeation may be partially derived from the differences on how widely the putative inner activation gate can open. We hope that our current findings can inspire future structural, computational, and functional studies to further establish the molecular mechanisms underlying $Ca^{2+}$-dependent substrate permeation of TMEM16 proteins.

## Methods

**Cell lines and cell culture**. HEK293T cells are authenticated and tested negative for mycoplasma by Duke Cell Culture Facility. The Cas9 control HEK293T and TMEM16F-KO HEK293T cell lines were generated by Duke Functional Genomics Core and were characterized in a recent report[31]. Briefly, the sgRNA (5′-AATAGTACTCACAAACTCCG-3′; see also Supplementary Table 1) targeting exon 2 of ANO6/TMEM16F was first cloned into lentiguide-puro (Addgene #52962). LentiCAS9-blast (Addgene #52962) was used to transduce HEK293T cells (Duke Functional Genomics Facility) to generate stable Cas9-expressing cells. All transductions were done at a multiplicity of infection (MOI) of <1 in the presence of 4 µg ml⁻¹ polybrene. Twenty-four hours post infection, cells were selected with 10 µg ml⁻¹ blasticidin or 2 µg ml⁻¹ puromycin for 48-72 h and then expanded. Genomic DNA was harvested from cells 1 week after transduction with sgRNA. The ANO6 exon 2 locus was PCR amplified from genomic DNA and subsequently analyzed using the Surveyor assay (Integrated DNA Technologies) to confirm the

presence of insertions and deletions (Supplementary Table 2). The TMEM16F-KO HEK293T cells were further purified using single-cell colonies from serial-dilution. The single-cell colonies were selected and expanded. Western blots and phospholipid scrambling assay were used to screen for the colonies without TMEM16F protein expression and function. The purified TMEM16F-KO HEK293T cells were used throughout to characterize phospholipid scrambling activity. HEK293T cell line with stable expression of C-terminally eGFP-tagged mTMEM16F was a kind gift from Dr. Min Li. All cells were maintained in Dulbecco's modified Eagle's medium (DMEM; Gibco, #11995-065) supplemented with 10% fetal bovine serum and 1% penicillin–streptomycin and supplied with 5% $CO_2$ at 37 °C. The same medium with 100 µg ml⁻¹ of hygromycin was used for culturing mTMEM16F stable HEK293T cells.

**Transfection**. pEGFP-N1 vector carries coding sequences of either murine TMEM16F (Open Biosystems cDNA # 6409332), murine TMEM16A (Open Biosystems cDNA # 30547439), or their mutations with C-terminal eGFP tag. The plasmids were transiently transfected to the TMEM16F-KO HEK 293T cells by using X-tremeGENE9 transfection reagent (Millipore-Sigma). Medium was changed 5-h post transfection with either regular (Gibco, #11995-065) or $Ca^{2+}$-free DMEM (Gibco, #21068-028). Experiments were proceeded 24–48 h after transfection. For constructs that induced spontaneous scrambling, experiments were done within 22–24 h post transfection.

**Microscope-based phospholipid scrambling assay**. Cells were seeded and transfected on poly-L-lysine (Sigma)-coated coverslips prior to the imaging experiments. Fluorescence-tagged (CF 594) Annexin V (Annexin V-CF 594, Biotium # 29011) was diluted at 0.5 µg ml⁻¹ with Annexin V-binding solution (10 mM HEPES, 140 mM NaCl, 2.5 mM $CaCl_2$, pH 7.4). The coverslips were placed into our homemade imaging chambers that contain the diluted Annexin V-CF 594. After focusing on the cell surface eGFP signal from the expressed TMEM16F-eGFP, an equal volume of Annexin V-CF 594 with 10 µM ionomycin, which was diluted from a fresh aliquot of 5 mM ionomycin (Alomone) stock solution, was added into the chamber to reach the final concentration of 5 µM to activate TMEM16F phospholipid scramblase. For Q-VD-OPh treatment, after TMEM16F-null HEK293T cells are transfected with eGFP-tagged mTMEM16F for 5-h, culture medium was replaced with the fresh medium containing 10 µM Q-VD-OPh (Sigma). After 24-h of transfection, the cells were tested for their scrambling activity as described above in the presence of 10 µM Q-VD-OPh.

In case of TMEM16A and its mutants, the final concentration of 2.5 µM of ionomycin was used. For Ani9 treatment, TMEM16A-L543K-expressing cells were treated with 10 µM Ani9, and then stimulated with 2.5 µM of ionomycin. The PS exposure of the cells was detected by the accumulation of Annexin V-CF 594 on the cells membrane. Time-lapse imaging of lipid scrambling activity was captured at 5-s intervals by a Prime 95B Scientific CMOS Camera (Photometrics) connected to an Olympus IX71 inverted epi-fluorescent microscope (Olympus IX73) for 10-min upon ionomycin application (0-min). A ×60 oil objective with NA of 1.35 was used for imaging. Image acquisition was controlled by Metafluo software (Molecular Devices).

**Quantifying phospholipid scrambling activity**. We used two ways to quantify TMEM16F lipid scrambling activity. For TMEM16F-expressing cells without spontaneous lipid scrambling activities prior to ionomycin application, $t_{1/2(Imax)}$ was analyzed. $t_{1/2(Imax)}$ indicates the amount of time each individual transfected cell takes to reach 50% of maximum AnV fluorescent intensity by the course of 10-min post ionomycin treatment. To determine $t_{1/2(Imax)}$, a customized MATLAB program (Mathworks) was written to analyze the AnV fluorescent intensity change overtime of each cell. Briefly, a region of interest (ROI) was manually chosen around the scrambling cells, and the AnV fluorescent intensity was calculated using the following equation for each frame:

$$I = \sum_{n=1}^{n=N} i \tag{1}$$

where $i$ equals the fluorescent intensity of each pixel and $N$ is the number of the pixels in the ROI. The time reaching half of $I_{max}$ was defined as $t_{1/2}$.

For the mutations that induce constitutive scrambling activity prior to ionomycin treatment, we quantify the percentage of spontaneous PS-positive cells by manually counting the number of AnV-positive expressing cells over all expressing cells, which were detected by GFP signals, in 10–20 random fields of view per coverslip. Each data point represents the percentage of spontaneous PS-positive expressing cells in one coverslip.

**Live cell Caspase 3/7-binding assay**. To detect cleaved Caspase 3/7 level in live cells, we employed Live cell Caspase 3/7 binding assay kit (AAT Bioquest, # 20101). After 24-h post transfection, culture media were removed and the cells were washed once with fresh media. The stock solution of TF3-DEVD-FMK (150×) was prepared and stored as described in the manual complemented with the assay kit (AAT Bioquest). The TF3-DEVD-FMK dye was diluted to 1× in either normal or $Ca^{2+}$-free medium depending on the mutations that were transfected to the cells.

After incubating with 1× dye for 1 h at 37 °C, 5% $CO_2$, the cells were washed twice with the kit's washing buffer. 1:100 dilution of CF 640R-tagged AnV in AnV-binding buffer (Biotium # 29014) was also added into the imaging chamber to monitor the cleaved caspase 3/7 and spontaneous PS exposure simultaneously. The results were collected with a ×63/1.4 NA Oil Plan-Apochromat DIC in Zeiss 780 inverted Confocal microscope. The results were analyzed with ImageJ and Zeiss.

**Electrophysiology**. TMEM16F-KO HEK293T cells were used for expression and recordings of mTMEM16F current. Normal HEK293T cells were used for expression and recordings of mTMEM16A current. HEK293T cells grown on poly-L-lysine (PLL, Sigma) and laminin (Sigma)-coated coverslips placed in a 24-well plate (Eppendorf) reaching 40–60% confluency were transiently transfected using X-tremeGENE 9 DNA transfection reagent (Sigma). Two hundred nanograms of DNA was used for each well to yield sufficient channel expression for recordings. All recordings were carried out at room temperature after 24–48 h following transfection. Inside-out recordings were performed on patches excised from cells expressing eGFP-tagged TMEM16A or TMEM16F constructs. Pipette electrodes were made from borosilicate glass capillaries (1.5 mm × 0.86 mm) with a puller (Sutter Instruments), fire-polished with a microforge (Narishge), and had a resistance of ~2–3 MΩ when filled with recording solutions. The pipette (external) solutions contain 140 mM NaCl, 5 mM EGTA, 10 mM HEPES, 2 mM $MgCl_2$, adjusted to pH 7.3 (with NaOH). The perfusion (internal) solution contain 140 mM NaCl, 10 mM HEPES, 5 mM EGTA, adjusted to pH 7.3 (with NaOH). The internal solution with 100 μM $Ca^{2+}$ was made by directly adding $CaCl_2$ into a solution containing 140 mM NaCl and 10 mM HEPES, adjusted to pH 7.3 (with NaOH). The internal solution with 2.26 μM $Ca^{2+}$ contains 140 mM NaCl, 10 mM HEPES, 5 mM EGTA, and 4.64 mM $CaCl_2$, adjusted to pH 7.3 (with NaOH). The amount of added $CaCl_2$ was calculated using WEBMAXC (https://somapp.ucdmc.ucdavis.edu/pharmacology/bers/maxchelator/). Free $Ca^{2+}$ concentration of EGTA-buffered solution was further verified using the ratiometric $Ca^{2+}$ dye Fura-2 (ATT Bioquest) and plotted against a standard curve from a calibration kit (Biotium).

For the current–voltage relationship (I–V) recordings, the membrane was held at −60 mV and 250-ms test voltage steps ranging from −120 mV to +140 mV were applied at a 20 mV increment. Due to the small amplitudes and rapid deactivation of the tail currents at the −60 mV repolarization step of TMEM16A in the absence of $Ca^{2+}$, steady-state peak currents from the test voltage steps were used to construct the I–V plot. For TMEM16F's I–V recordings in the presence of 2.26 μM and 100 μM $Ca^{2+}$, peak tail currents measured at −60 mV repolarization steps were normalized to the max tail current to generate the G–V curves.

For the reversal potential ($E_{rev}$) measurements, TMEM16A and TMEM16F channels were first activated by perfusion of internal solution containing 100 μM $Ca^{2+}$. Both the pipette (external) and bath (internal) are symmetric and contain 140 mM NaCl, 5 mM EGTA, 10 mM HEPES, adjusted to pH 7.3 (NaOH). For TMEM16A, a ramp protocol from −100 mV to +100 mV was used to elicit currents. For TMEM16F, an inverted V-shaped protocol in which the membrane was ramped from −120 mV to +120 mV and back to −120 mV was used. The second phase of the ramp (from +120 mV to −120 mV) was used to construct the I–V plots for measuring the $E_{rev}$. For both TMEM16A and TMEM16F, the changes in $E_{rev}$ were triggered by switching internal solution with a low NaCl solution containing 14 mM NaCl, 5 mM EGTA, 10 mM HEPES, and 100 μM $Ca^{2+}$ adjusted to pH 7.3. The $E_{rev}$ was determined as the membrane potential at which the current was zero, and the shifts in the $E_{rev}$ ($\Delta E_{rev}$) are the differences between the $E_{rev}$ measured in symmetric NaCl and low internal NaCl. The permeability ratio $P_{Cl}$/$P_{Na}$ was calculated using the Goldman–Hodgkin–Katz equation:

$$V_m = \frac{RT}{F} \ln \frac{P_{Na}[Na]_o + P_{Cl}[Cl]_i}{P_{Na}[Na]_i + P_{Cl}[Cl]_o} \qquad (2)$$

in which $V_m$ is the measured $E_{rev}$ shift ($\Delta E_{rev}$); $P_{Na}$ and $P_{Cl}$ are the relative permeabilities of $Na^+$ and $Cl^-$; $[Na]_o$ and $[Na]_i$ are external and internal $Na^+$ concentrations; $[Cl]_o$ and $[Cl]_i$ are external and internal $Cl^-$ concentrations; $F$ is the Faraday's constant (96485 C mol$^{-1}$); $R$ is the gas constant (8.314 J mol$^{-1}$), and $T$ is the absolute temperature (298.15 K or 25 °C).

For time-course monitoring of TMEM16F channel activity, a voltage step protocol in which the membrane was stepped to +80 mV and then −80 mV lasting 200 ms each. The membrane was held at 0 mV. Channel opening was elicited by perfusion of internal solution containing 100 μM $Ca^{2+}$. Peak currents measured at +80 mV steps were used for time-course monitoring of channel opening.

All electrophysiology recordings were low-pass filtered at 5 kHz (Axopatch 200B) and sampled at 10 kHz (Axon Digidata 1550 A) and digitized by Clampex 10 (Molecular Devices). Offline data analysis was performed in Clampfit, Excel (Microsoft), and Prism (GraphPad).

**Sequence alignment**. Protein sequences of nhTMEM16, afTMEM16, murine (m) TMEM16A (UniProtKB: Q8BHY3), mTMEM16B (UniProtKB: Q8CFW1), mTMEM16C (UniProtKB: A2AHL1), mTMEM16D (UniProtKB: Q8C5H1), mTMEM16E (UniProtKB: Q75UR0), mTMEM16F (UniProtKB: Q6P9J9), mTMEM16G (UniProtKB: Q14AT5), mTMEM16H (UniProtKB: Q6PB70), mTMEM16J (UniProtKB: P86044), and mTMEM16K (UniProtKB: Q8BHY79) were aligned using Clustal Omega (https://www.ebi.ac.uk/Tools/msa/clustalo/).

**Homology modeling of TMEM16F in open and closed states**. Homology models of the closed (without $Ca^{2+}$) and intermediate (with $Ca^{2+}$) states of mTMEM16F were derived directly using $Ca^{2+}$-free (PDB: 5OYG) and $Ca^{2+}$-bound (PDB: 5OYB) mTMEM16A structures respectively with the sequence alignment shown in Supplementary Fig. 1a using the Swiss-pdb server[39]. Even though the $Ca^{2+}$-bound structure of nhTMEM16 (PDB: 4WIS) is believed to provide a good model of the open conformation of the TM region of mTMEM16F, the sequence conservation in the extracellular domain and loops is poor. To derive a model of mTMEM16F in the open state, steered MD was initiated from the intermediate state model derived from 5OYB to reposition TMs 3–6 to the configuration observed in the $Ca^{2+}$-bound nhTMEM16 structure (PDB: 4WIS) using the CHARMM software[40,41]. We note that, even though TM5 is similarly positioned between PDBs 5OYB and 4WIS, it lines the putative phospholipid permeation pathway and thus was included in the steered MD to ensure the structural integrity. For this, a homology model of mTMEM16F TM region alone was first derived from nhTMEM16 using the sequence alignment derived from Clustal Omega (see above). This model was then used as the target (open state TM conformation) during the steered MD simulation, where root mean square deviation (RMSD) restraint was applied on TMs 3–6 with increasing force constants ranging from 5 to 75 kcal mol$^{-1}$ Å$^{-2}$. In addition, all heavy atoms of adjacent five residues of loops before and after TMs 3–6 were harmonically restrained using a weak force constant of 0.1 kcal mol$^{-1}$ Å$^{-2}$. This allows appropriate, well-controlled loop reconfiguration to accommodate the movement of TMs 3–6. The rest of the whole protein were restrained by harmonic positional restrains with a force constant of 100 kcal mol$^{-1}$ Å$^{-2}$ to avoid unnecessary structure disruption. The simulation was performed in vacuum with a distance-dependent dielectric constant. Timestep was set to 1 fs. The final model was only ~0.14 Å backbone RMSD from 4WIS for TMs 3–6 (Supplementary Movie 1).

**MD simulations of mTMEM16F**. The mTMEM16F structures (in open, intermediate, or closed state) were first inserted in model lipid bilayers and then solvated in TIP3P water[42] using the CHARMM-GUI web server[43]. To mimic mammal plasma membranes, pure POPC and 2:1 POPC/POPS mixture were used for upper and lower leaflets, respectively. The solvated systems were then neutralized and 150 mM KCl was added. The final simulation boxes contain about ~660 lipid molecules and ~73,000 water molecules, with a total of ~335,000 atoms and dimensions of ~155 × 155 × 140 Å$^3$. The CHARMM36m all-atom force field[44] and the CHARMM36 lipid force field[45] were used. All simulations were performed using CUDA-enabled Amber14[46]. Long-range electrostatic interactions were described by the Particle Mesh Ewald (PME) algorithm[47] with a cutoff of 12 Å. Van der Waals interactions were cutoff at 12 Å with a smooth switching function starting at 10 Å. The lengths of hydrogen-containing covalent bonds were constrained using SHAKE[48] and the MD time step was set at 2 fs. The temperature was maintained at 298 K using the Langevin dynamics with a friction coefficient of 1 ps$^{-1}$. The pressure was maintained semi-isotropically at 1 bar at both x and y (membrane lateral) directions using the Monte Carlo (MC) barostat method. Several segments were absent in the cryo-EM structures of TMEM16A; they were considered dynamic and thus not included in simulations. These include the N- and C-terminal segments (M1-N86 and S882-E911) and a long loop in the cytosolic domain (Y101-V134). The residues before and after the missing part were capped with either an acetyl group (for N-termini) or a N-methyl amide (for C-termini). To minimize the effects of missing residues on the cytosolic domain, the backbone of structured parts of the cytosolic domain (E92-A100, L135-F182, S190-D194, P206-V220, G232-L248, R420-V431, K862-E881) were harmonically restrained with a modest force constant of 1 kcal mol$^{-1}$ Å$^{-2}$ during all simulations.

All systems were first minimized for 5000 steps using the steepest descent algorithm, followed by a series of equilibration steps where the positions of heavy atoms of the protein/lipid were harmonically restrained with restrained force constants gradually decreased from 10 to 0.1 kcal mol$^{-1}$ Å$^{-2}$. In the last equilibration step, only protein heavy atoms were harmonically restrained and the system was equilibrated 10 ns in under NPT (constant particle number, pressure, and temperature) conditions. All production simulations were performed under NPT conditions. For each state (open, intermediate, and closed), three independent 400-ns simulations were performed. Snapshots were saved every 50 ps. Only snapshots from 100 to 400 ns of each trajectory were included for analysis. The PMFs of water and phosphate group of lipids were calculated directly from the corresponding probability distributions along the membrane normal. The standard errors were estimated from the calculated averages over three independent simulations. The 3D phosphate density distribution near the channel was calculated using a cubic grid with a resolution of 0.5 Å.

**Statistical analysis**. All statistical analyses were performed in Prism software (GraphPad). Unpaired two-tailed Student's t-test was used for single comparisons between two groups, and one-way ANOVA (with Tukey's multiple comparisons test) was used for multiple comparisons. Unless otherwise described, the data are representative of mean ± standard error of the mean (SEM). P values less than 0.05 were considered statistically significant.

**Reporting summary**. Further information on research design is available in the Nature Research Reporting Summary linked to this article.

## Data availability

Data supporting the findings of this manuscript are available from the corresponding author upon reasonable request. A reporting summary for this Article is available as a Supplementary Information file. The source data underlying Figs. 2b, 2c, 3b, 3d, 4b, 5a–c, 6b, 6c, 6e and Supplementary Figs. 3b, 4c, 4d, 4e, 4f, 4i, 4j, 7f are provided as a Source Data file.

## Code availability

Code for quantification of CaPLSase activity using fluorescence microscopy is available at Github (yanghuanghe/scrambling_activity).

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

## Acknowledgements

We are grateful to So Young Kim from Duke Functional Genomics Core for the assistance on generating the TMEM16F HEK293T knockout cell line using CRISPR-Cas9 and Min Li for generating the TMEM16F stable HEK293T cell line. This work was supported by grants NIH-DP2GM126898 (to H.Y.), R00-NS086916 (to H.Y.), the CTHD Young Investigator Translational Research Award (to H.Y.) under grant U54-HL112307 (B.A.S. and R.C.B.), the Whitehead Foundation (to H.Y.), and NIH-R01-HL142301 (to J.C.).

## Author contributions

H.Y. conceived and supervised the project. T.L. performed mutagenesis, phospholipid scrambling experiments, and data analysis. Z.J. performed computational experiments under the guidance of J.C. S.C.L. performed electrophysiology experiments. Y.Z. wrote scripts for imaging analysis. H.Y., J.C., T.L., S.C.L., and Z.J. wrote the manuscript.

## Additional information

**Competing interests:** The authors declare no competing interests.

