## [Peer Review File · Nature Communications]

Reviewers' Comments:

Reviewer #2:

Remarks to the Author:

Here the authors explore the lipid scrambling properties of TMEM16F, and to a lesser extent the ion channel TMEM16A, using a combination of cell based assays, mutagenesis, and computer simulation carried out on homology models. These proteins are of extreme interest to researchers since they play roles in a wide range of biological phenomena, but they are also fascinating machines as they are ion channels and lipid scramblases, and it is not entirely clear how these proteins accomplish these tasks.

Here, the authors identify three residues in TMEM16F near the center of the hydrophilic groove that play a crucial role in lipid gating: F518, Y563, and I612. The data presented showing the role of these residues (and mutants at these sites) is very compelling. Impressively, they show that F518K and Y563K make the channel Ca⁺⁺ independent scramblases that are constitutively open. Moreover, the single point mutation in TMEM16A of L543K turned this ion channel into a lipid scramblase. This is impressive, as previous work on chimera structures by the Hartzell lab narrowed down the scramblase domain to a stretch of 11-12 residues, this is a single point mutation.

Simulations

While I have questions regarding the experiments below, my primary concerns center on what can be said, and what should be said, from the simulations. I find some of the simulations compelling and interesting, but in many cases, free energy values should not be computed since you are doing all of your simulations on homology models or homology models that you then drive to new states with SMD. Additionally, your simulations are all rather short (400 ns or so) to present free energy values (more on this later).

The authors use the Ca⁺⁺-free and Ca⁺⁺-bound TMEM16A structure to construct "closed" models of TMEM16F (PDB ID: 5OYG & 5OYB). They then simulate these homology models and see that that the Ca-bound model shows TM6 straighten and makes the vestibule more accessible to water, but there is still a major constriction site.

The authors created an open state of TMEM16F by using steered MD to open the groove at the constriction site of 3,4,6. Why use steered MD to reposition TM3-6 rather than making a homology model on nhTMEM16? How did you steer the RMSD with respect to 4WIS? Did you make an alignment to do this, or did you somehow do this in a sequence agnostic manner? What are the residues at the SE site doing during this transition? Why did you do the steered md in vacuum? Overall, I am highly skeptical of this approach, and there is the potential for introducing lots of strain into the structure with lots of high energy distortions.

Simulations of the homology models are then used to compute free energy profiles for water and phosphate movement through the groove. This is based on 100 ns of unbiased simulation determined from densities. These are very short simulations for carrying out such analysis, and given the fact that you are using homology models, I encourage the authors not to report energy values – I would not believe them at all given the small sampling coupled with homology models. I would discuss qualitative features of the simulations of the models and show density profiles, but computing energies (which is always possible) lends a kind of quantification and accuracy to what you have done that simply is not there.

The authors state that they observe multiple lipid permeation events. How many 2-4? Are these PC and PS?

On lines 200-202 the authors discuss calculating the free energy difference for channel opening on the WT and F518K models. First I worry that your states are not correct, or even if they are close,

this is a homology model. Second, even if you were starting from crystal structures, these are very hard calculations to converge. I can't tell exactly how many bins you used (0.2 Å spacing going between 1.4 to 3.6 Å RMSD or maybe 0 to 3.6 Å). In the later case, you have done 20 ns/bin x 20 or so bins ~ 400 ns aggregate simulation. 400 ns is not enough time for this high dimensional system to converge. The authors are claiming that the energy difference between the two models is ~ 1 kcal/mol. This is the error we might hope for in a small simulation that is very well sampled (many microseconds to milliseconds), and TMEM16 simulations are enormous by those standards. For instance, read the lipid bilayer free energy calculations for indolicidin by Regis Pomes (a leading expert in free energy calculations and membrane protein simulations) (Neale et al. Biophysical Journal 106:8 P. L29-L31). The authors argue that lipid relaxation around the protein occurs on the microsecond timescale, and simulating less than that in some bins influences the PMF. Thus, I would not show your free energy profiles.

Experiments

On page 6, the authors introduce their experimental procedure. They start talking about fluorescence for the HEK cell assays, but they don't say what they are measuring. The authors use $t_{1/2}$ values to determine how active a mutant is, but what about the total fluorescent change? Is there a strong correlation between greater fluorescence and smaller $t_{1/2}$ values and less fluorescence and greater $t_{1/2}$ values?

What does the D703R loss of function mutation do when coupled with F518K or Y563K or (D703R/F518A for that matter)? Based on the authors claims, we would expect that this double mutant would still be a constitutive scramblase.

The triple mutant was a good idea and it is very active at PS scrambling. Does the Ala triple mutant have to gate?? The authors could have taken the closed, Ca⁺⁺ TMEM16A structure, make the triple mutant (don't force TM helices apart) and look at whether they see permeation. This is important. What is your model telling you about restudies blocking lipid passage versus biasing the channel to different functional states. Ala is interesting too as it is still hydrophobic. I like the bulky residue work to show reduced PS exposure; however, the F518W is still opposite of what might be expected, and the authors haven't addressed whether the bulky residues are pushing the channels to different states (closed versus open) or simply making it impossible for the open state to permeate (unless I missed it). Along these lines the authors say F518L is a LOF, but isn't F bigger and more hydrophobic than L? This confused me some.

As stated on lines 180-181 – all mutations at the inner gate/neck region caused GOF. Interesting, and not unexpected.

On line 213 page 10, these mutants from the Tien paper that destroy Ca⁺⁺ binding also abolish ion channel activity, right? If so, please mention that. I do like that the authors show that the single point GOF mutants (except two) can't facilitate scrambling when the Ca⁺⁺ site is knocked out suggesting that the GOF mutants can't gate the channel on its own, and possibly hinting that the closed state of the channel can't necessarily be made to flip lipids simply by changing a residue in the pathway. That said, the bulky, aromatic residues F518 and Y563 can flip without Ca⁺⁺ when a lysine is introduced. Are these scramblases still gating like normal or not? This is interesting. Is it the sign of the charge that matters (because the assay is testing PS flipping) or simply the charge? Double mutants with D703R+F518D/E/R and D703R+Y563D/E/R, would be very interesting.

With regard to the TMEM16A mutations discussed on lines 235-238 please let us know here if L543 and I637 are the homologous sites to F518 and Y563. Also, please indicate what you mean by opening in the text: Cl⁻ channel activity or lipid scrambling?

No where in the paper can I find the word "serine" except in the references. On line 138, the authors discuss PS exposure – what the assay is detecting – but it isn't discussed in detail. The

simulations probe PC movement, but PS is the lipid of interesting. There is a lot of difference in the lipid types when it comes to gating, selectivity, and function (the Accardi group has nicely shown this in several papers). The authors should clearly point this out so that the readers appreciate this.

Minor Points

The authors state that the SE site charged residues are not conserved – this isn't true is it? They may have a charge reversal, but they exist in almost all family members, right?

Figure 5 caption mentions WT in panel A, but I don't think WT is shown.

On lines 278-279 the authors state that the motions are centered on TM6. However, doesn't it appear that TM4 and TM3 are just as big when we compare different solved structures from different subfamilies?

Reviewer #3:

Remarks to the Author:

In this report, the authors introduced mutations in several amino acids of mouse TMEM16F that functions as a phospholipid scramblase, and propose a model how this protein transport phospholipids between the two leaflets of plasma membranes. I appreciate the effort to characterize this interesting membrane protein. But unfortunately, the data are preliminary, and are not well performed. My comments are as follows.

1. The scramblase activity (the exposure of phosphatidylserine, PS) of the TMEM16F derivatives was assayed by transient-over expression in 293T cells. The authors often observed caspase-positive cells in the Ca^{2+} -ionophore treated transfectants. This is not a good experiment, or it is a dirty experiment. As the authors are aware, PS is exposed in apoptotic or necrotic cells. The authors argue that they examined the PS-exposure in caspase-negative cells. But, we do not know how much caspase should be activated to induce PS-exposure, and how the caspase-detecting system is sensitive. To study the PS-exposure by TMEM16 family member, it is essential to study with the stable transformants. Please see reference 5, and Gyobu et al. (PNAS 114, 6274, 2017).
2. With the data obtained with the mutants, the authors discuss the detailed scrambling mechanism, and propose that Ca^{2+} causes the conformational change from Open to Closed structure (Figure 6). Recently, Bushell et al (<http://dx.doi.org/10.1101/447417>) determined the tertiary structure of Ca^{2+} -bound and unbound forms of TMEM16K, a homolog of TMEM16F that scrambles phospholipids in Ca^{2+} -dependent manner. They report that both Open and Closed conformation can be found in Ca^{2+} -bound protein. Thus, the proposal in this manuscript is mis-leading.

Reviewer #4:

Remarks to the Author:

Review Manuscript by Le et al. entitled "An inner activation gate controls TMEM16F phospholipid scrambling"

In this manuscript Le et al. investigate the lipid permeation pathway in the Ca-activated scramblase TMEM16F by using mutagenesis, scramblase activity assays, electrophysiology and MD simulations.

As template to guide the modeling and the selection of the residues to be mutated, they used the structures of Ca-bound and Ca-free mouse TMEM16A and of the Ca-bound fungal nhTMEM16.

The Authors obtained two main results.

First they identified as inner gate for lipid transport three hydrophobic residues, F518, Y563 and I612, in the middle of the phospholipid permeation pathway. Interestingly a single mutation of one of these aminoacids can disrupt the gate, in some cases leading to a constitutively active scramblase.

Second they converted the Ca-activated chloride channel TMEM16A into a constitutively active scramblase by a single mutation of one of the corresponding aminoacids (L543K) identified in TMEM16F. This second result is really surprisingly. The overall similarity of all TMEM16 is not a surprise and Hartzell's and Nagata's groups (Yu et al 2015, Gyobu et al 2016, Gyobu et al 2017) converted TMEM16A into a scramblase, but by introducing a stretch of 35 aminoacids of a TMEM16 scramblase into TMEM16A!

The results are interesting, but more experiments are needed to be completely convincing.

Major

1) In most TMEM16s scrambling activity has been associated to little selective ionic current (TMEM16F Yang et al 2012; TMEM16E Dizanni et al 2018; nhTMEM16 Lee et al 2016; aTMEM Malvezzi et al 2013). It makes exception the recently identified *Dictyostelium discoideum* TMEM16 (Pelz et al 2018), but in this case very limited data are available. Additionally gain of function mutations in scrambling activity have been associated to gain of function current activity both in TMEM16E (see Fig. 6 Di Zanni et al 2018) and TMEM16F (D409G, see Fig9 Scudieri et al, J Physiol 2015). Therefore comes up very naturally the question "what happens to the current of TMEM16F with the mutations described in this manuscript?" Are the gain of function mutations for scrambling activity of TMEM16F also showing a gain of function in current activity? I do not ask for a full characterization of all mutations, but I think it will complete the work to show whether there is a gain of function activity at least for the most prominent ones. Possibly use TMEM16F-KO HEK cells.

2) Page 7 A cytotoxic effect due to constitutive scrambling activity in cells expressing mutant TMEM16F is described. It is probably supporting your data the paper by Dizanni et al 2018 where it is reported a rounded cell morphology when it is expressed a gain of function mutant of TMEM16E. Another gain of function mutation has been previously described for TMEM16F (D409G) and used in this work. Have you observed also here a cytotoxic effect? Reference for D409G mutant is missing.

3) The conversion of TMEM16A CaCC to a scramblase with a single mutation is a key point. I think that further evidence should be given to support this surprising result

a. Please show in Figure 5 also cells expressing TMEM16A-L543K showing cytotoxicity.

b. Is An-V assay performed in TMEM16F KO cells? (specify in the legend) If not it should be done.

c. Can scramblase activity of TMEM16A L543K blocked by TMEM16A blockers such Ani9, MONNA or CaCCinh01?

d. Do in excised patches currents mediated by TMEM16A-L543K run down as WT?

e. Fig 5a. Why there are not tail currents upon repolarization? Do you see them with WT (with Ca)? In symmetric chloride I would expect an inward current upon repolarization. Is selectivity Cl/cations affected for L543K mutant? In Yu et al 2015 chimeric constructs TMEM16A/TMEM16F show a change in selectivity.

f. TMEM16A mutant I637K does not show scrambling activity, but only current activity. It should be discussed. Is mutant S588K showing current in high Ca?

4) Student's test was used, however to compare multiple data is not appropriate. Anova and an appropriate post-hoc test should be used.

Medium

1. It is missing in the main manuscript a figure with a representative experiment for scrambling activity of wild type and a gain of function mutant TMEM16F. Reference to supplementary figure and to another paper of the same authors is given (submitted to JBC), however it would be better to have in the main text at least a graph of total fluorescence vs time, from which you estimate $t_{1/2}$, to appreciate the validity of your analysis on TMEM16F mutants (as in Fig4 of the submitted paper and in Supplementary figure 4). Additionally the value of the $t_{1/2}$ includes the wash-in of the solution and by the time necessary for enough annexin-V binding to PtdSer. It should be

- evaluated. For example from the D409G gain of function mutation, that scrambles without ionophore (in the supporting materials also ionomycin is applied) or from apoptotic cells.
2. It should be specified that Annexin V binding assay and ionomycin application is made in presence of Ca even when cells have been kept in Ca-free medium to avoid cytotoxicity (for example page 11).
 3. Page 12 Electrophysiology experiments are not properly described in the results
 4. Specify in each legend if TMEM16F KO cells have been used or HEK cells

Minor

I do not find necessary to use acronyms for gain of function and loss of function (as pure suggestion I would not use GOF and LOF).

CaPLSase stands for Ca-activated phospholipid scramblase. Therefore authors should pay attention whether to use PLSase instead of CaPLSase. For example in the text they often report constitutively active CaPLSase, shouldn't it be PLSase?. In some mutants the Ca-dependence is actually lost.

Page 3 line 44. PGD is introduced but never used in the text

Page 4 line 73, 14 line 294 CaCC channel .

Page 6 Add somewhere that you used an annexin-V based assay.

Page 7 line 138. Specify that the other mutants do not show activity in absence of ionophore.

Page 13 line 268, specify that corresponds to F518 in TMEM16F, as reported later line 293

Pag 13,line 280 groove instead of grove

Page 13,line 282, add ref to the 'credit card mechanism'

Pag 18 line 384 manually

Supplementary figures are referred to as Supplementary/Supplemental Fig 2 or Supplementary Fig S2

Figures

Legends in figure panels are too small

Fig 2b,c, fig3b, Fig4b, fig 5e Reduce the number of ticks with numbers in the Y axis

Fig 2b Xaxis labels are hidden by the panel below and the right part is missing

Fig3b, 5e histogram is missing for some mutations. If it is done because too few cells scramble, it should be written in the legend.

Fig 4b For the constitutively activated TMEM16F mutants that caused >80% of spontaneous PS exposure add at least the pie graph.

Fig 5 time and current scale is missing.

Fig5b increase height of the panel

Fig S2a letters are too small (and stretched). Difficult to read. Black is too faint (at video)

Legends

Fig1 line 632 TMEM16F

Fig4b Although cells were kept ion Ca-free medium, when ionomycin is applied Ca is present.

Specify it in the legend

Fig 5, specify when ionomycin was used.

Fig 5d specify that representative images of L543K-TMEM16A-expressing cells are from cells kept in Ca-free medium-

In some panels is missing the number of experiments. I would add it to the legend.

Fig. S8 I assume that legend is wrong. Specify that at t=0min there is no annexin (otherwise I would expect staining of L543K mutant)

Dear Reviewers:

We appreciate your critical comments and constructive criticisms, which were very helpful in improving our manuscript.

We have completed additional experiments and substantially revised the manuscript based on the reviewers' suggestions. The major revisions include: 1) clarifications of our rationales of building the TMEM16F homology models and computational procedures; 2) removal of the free energy calculations of open-closed transition for WT and F518K TMEM16F (previously Fig. 4e); 3) addition of a new section to the Results to describe the scrambling assay and our quantification in details; 4) more experiments to demonstrate that transient transfection of TMEM16F into HEK293T cells, which does not induce apoptosis, is reliable to measure scrambling activity; and 5) electrophysiological recordings to characterize the gain-of-function inner gate mutations in TMEM16F and TMEM16A. Please see our point-to-point responses below for details.

As we have done significant revision on the manuscript to improve its quality, it is difficult to use the Word 'track changes' function to track our editing. Instead, we submit a merged pdf file of our original submission combined both manuscript and supplementary figures as a supporting document. We apologize for the inconvenience.

Reviewer #2 (Remarks to the Author):

Here the authors explore the lipid scrambling properties of TMEM16F, and to a lesser extent the ion channel TMEM16A, using a combination of cell based assays, mutagenesis, and computer simulation carried out on homology models. These proteins are of extreme interest to researchers since they play roles in a wide range of biological phenomena, but they are also fascinating machines as they are ion channels and lipid scramblases, and it is not entirely clear how these proteins accomplish these tasks.

Here, the authors identify three residues in TMEM16F near the center of the hydrophilic groove that play a crucial role in lipid gating: F518, Y563, and I612. The data presented showing the role of these residues (and mutants at these sites) is very compelling. Impressively, they show that F518K and Y563K make the channel Ca⁺⁺ independent scramblases that are constitutively open. Moreover, the single point mutation in TMEM16A of L543K turned this ion channel into a lipid scramblase. This is impressive, as previous work on chimera structures by the Hartzell lab narrowed down the scramblase domain to a stretch of 11-12 residues, this is a single point mutation.

Response:

We really appreciate the positive comments. Detailed responses are listed below.

Simulations

While I have questions regarding the experiments below, my primary concerns center on what can be said, and what should be said, from the simulations. I find some of the simulations compelling and interesting, but in many cases, free energy values should not be computed since you are doing all of your simulations on homology models or homology models that you then drive to new states with SMD. Additionally, your

simulations are all rather short (400 ns or so) to present free energy values (more on this later).

The authors use the Ca⁺⁺-free and Ca⁺⁺-bound TMEM16A structure to construct “closed” models of TMEM16F (PDB ID: 5OYG & 5OYB). They then simulate these homology models and see that that the Ca-bound model shows TM6 straighten and makes the vestibule more accessible to water, but there is still a major constriction site.

The authors created an open state of TMEM16F by using steered MD to open the groove at the constriction site of 3,4,6. Why use steered MD to reposition TM3-6 rather than making a homology model on nhTMEM16? How did you steer the RMSD with respect to 4WIS? Did you make an alignment to do this, or did you somehow do this in a sequence agnostic manner? What are the residues at the SE site doing during this transition? Why did you do the steered md in vacuum? Overall, I am highly skeptical of this approach, and there is the potential for introducing lots of strain into the structure with lots of high energy distortions.

Response: We agree with the reviewer that caution needs to be taken when utilizing the computational tools and interpreting the computational results. The reason that we did not directly construct the open state model using the nhTMEM16 structure was due to the low sequence identity between nhTMEM16 and mTMEM16F with the exception of some TM segments (Supplementary Fig. 1a). In particular, the extracellular loops of mTMEM16F are largely absent in nhTMEM16. Even within the TM region, the structures of nhTMEM16 and mTMEM16A show significant conformational differences in TMs 3, 4 and 6 and to a lesser extent in TM5. By contrast, TMEM16A and TMEM16F share ~35% in overall sequence identity and even higher sequence identity in their TM domains. Therefore, we believe that the mTMEM16A structures could serve as suitable templates for modeling the overall structure of mTMEM16F, while the Ca²⁺-bound nhTMEM16 structure provides a reasonable template for modeling the open conformation of the TM region (particularly that of TMs 3-6). We revised our manuscript accordingly to clarify our rationales (Page 5).

The steered MD protocol was designed specifically for TMs 3-6. We first created a homology model of the TM region of mTMEM16F based on the Ca²⁺-bound nhTMEM16 structure (PDB: 4WIS) using the Swiss-pdb server and sequence alignment derived from Clustal Omega. The TM region from the resulting TMEM16F model was then used as the target for the steered MD simulation. The steered MD was used here mainly as a flexible structure fitting procedure and therefore a simplified treatment of electrostatics using distance-dependent dielectric constant (RDIE) with full vdW and other interactions should be adequate. We note that TMs 3-6 line the hydrophilic groove and are solvent accessible. RDIE is actually a reasonable “implicit solvent” model and has been widely used in protein and ligand docking protocols. Nonetheless, we fully agree with the risk of introducing strains into the structure. Thus, we took cautious measures to minimize introducing structural artifacts when designing our steer MD protocol. For example, all heavy atoms of adjacent 5 residues of loops before and after the TM helices were also harmonically restrained using a weak force constant of 0.1 kcal/mol/Å². This allows

appropriate, well-controlled loop reconfiguration to accommodate the movement of TMs 3-6. For TMs 3-6, RMSD restraints were imposed with respect to the target conformation using gradually increasing force constants (from 5 to 75 kcal/mol/Å²) during multiple cycles. In addition, the rest of the protein was harmonically restrained with a large force constant of 100 kcal/mol/Å² during steer MD. Subsequent energy minimization and equilibration in explicit membrane and water also confirmed that the resulting structure of the open state does not seem to contain any high energy distortions.

We have added a statement clarifying these issues in the main text (Page 5 and 6). We also provided additional details of the steered MD protocol in Materials and Methods (see “Homology modeling of TMEM16F in open and closed states”).

A series of elegant studies from multiple labs have clearly demonstrated the importance of the S_E site in controlling lipid scrambling of the fungal and mammalian TMEM16 scramblases. However, when we performed sequence alignment and compared the available structures of the fungal nhTMEM16 and the mouse TMEM16A, we noticed that the composition of the S_E site in mammalian TMEM16 scramblases may not be exactly the same as that in nhTMEM16, even if a charge reversal is taken into account (Response Fig. 1). First, the equivalent S_E residue Arg residue in TM3 of TMEM16A is far away from the fungal S_E site Glu residues by sequence (Response Fig. 1a). Second, this Arg (R515 in mTMEM16A or R478 in mTMEM16F) is about 11 Å away from the TM6 acidic residue (E604 in mTMEM16F or E633 in mTMEM16A), which is equivalent to Arg432 of the fungal S_E site based on the recent TMEM16A structures (PDB: 5OYB and 5OYG) (Response Fig. 1c,d). Based on these observations, we're tempted to speculate that the mammalian TMEM16 S_E sites could arrange differently from the nhTMEM16 S_E site. As we are uncertain about which residues form the S_E site in TMEM16F and we lack structural and functional evidence to further support our hypothesis, we are hesitate to interpret our simulation on S_E site. Since S_E site is not the focus of this study, we have removed the discussion on the conservation of S_E site in the revised manuscript, as well as the original Supplementary Fig. 1 (which is now Response Fig. 1).

Response Figure 1. The reported S_E residues that control phospholipid permeation through the fungal TMEM16 CaPLSases may not be conserved in mammalian TMEM16 CaPLSases. **a**, Sequence alignment of the fungal and murine TMEM16 proteins. Transmembrane (TM) segments are highlighted in grey. The S_E site residues that were proposed to involve in extracellular gating are colored in red and blue for negative and positive charges, respectively. The top and bottom numberings are based on nhTMEM16 and mTMEM16F, respectively. **b**, The residue numbers of the proposed S_E site residues in nhTMEM16, mTMEM16A and mTMEM16F. **c**, The network of charged residues of nhTMEM16 S_E site (PDB: 4WIS). **d**, The ‘equivalent’ S_E site residues in the Ca²⁺-bound TMEM16A structure (PDB: 5OYB).

Simulations of the homology models are then used to compute free energy profiles for water and phosphate movement through the groove. This is based on 100 ns of unbiased simulation determined from densities. These are very short simulations for carrying out such analysis, and given the fact that you are using homology models, I encourage the authors not to report energy values – I would not believe them at all given the small sampling coupled with homology models. I would discuss qualitative features of the simulations of the models and show density profiles, but computing energies (which is always possible) lends a kind of quantification and accuracy to what you have done that simply is not there.

Reponses: While we agree that 400 ns (not 100 ns) unbiased simulations are far from adequate to sufficiently sample the high free energy regions, which will show no density (and hence infinite free energy). However, we respectfully argue that the density profiles that the reviewer suggested are actually equivalent to the “free energy” profiles shown in the revised Fig. 1d. The free energy profile is essentially a density profile in a log scale, which we believe provides a more intuitive and interpretable presentation.

The authors state that they observe multiple lipid permeation events. How many 2-4? Are these PC and PS?

Reponses: We have observed a total of three lipid permeation events (one in each trajectory). All three events involved PC instead of PS, likely a direct consequence of the ratio of 2:1 POPC/POPS mixture used in our simulations.

On lines 200-202 the authors discuss calculating the free energy difference for channel opening on the WT and F518K models. First I worry that your states are not correct, or even if they are close, this is a homology model. Second, even if you were starting from crystal structures, these are very hard calculations to converge. I can't tell exactly how many bins you used (0.2 Å spacing going between 1.4 to 3.6 Å RMSD or maybe 0 to 3.6 Å). In the later case, you have done 20 ns/bin x 20 or so bins ~ 400 ns aggregate simulation. 400 ns is not enough time for this high dimensional system to converge. The authors are claiming that the energy difference between the two models is ~ 1 kcal/mol. This is the error we might hope for in a small simulation that is very well sampled (many microseconds to milliseconds), and TMEM16 simulations are enormous by those standards. For instance, read the lipid bilayer free energy calculations for indolicidin by Regis Pomes (a leading expert in free energy calculations and membrane protein simulations) (Neale et al. Biophysical Journal 106:8 P. L29-L31). The authors argue that lipid relaxation around the protein occurs on the microsecond timescale, and simulating less than that in some bins influences the PMF. Thus, I would not show your free energy profiles.

Reponses: We totally agree that achieving adequate convergence in the current case is not feasible, due to the exact reasons detailed by the reviewer. We agree that the free energy profiles are not reliable and can be misleading. Given that these simulations are not essential to the main conclusions of this work, we agree with the reviewer's recommendation and have completely removed these results from this revised manuscript.

Experiments

On page 6, the authors introduce their experimental procedure. They start talking about fluorescence for the HEK cell assays, but they don't say what they are measuring. The authors use $t_{1/2}$ values to determine how active a mutant is, but what about the total fluorescent change? Is there a strong correlation between greater fluorescence and smaller $t_{1/2}$ values and less fluorescence and greater $t_{1/2}$ values?

Reponses: We apologize for the confusion. Since we described our lipid scrambling assay in details in the accompanied manuscript (now in JBC, Le, et al, *JBC*, 2019. doi:10.1074/jbc.AC118.006530), we did not do a good job explaining this assay clearly in the previous version of the manuscript. Now, in addition to citing this paper, we have moved the original Supplementary Fig. 3 to the main text as the new Fig. 2, and added a new section "Application of an optimized Annexin V (AnV)-based imaging assay to quantify CaPLSase activity" in the Results section to introduce the detailed procedure and our quantification method (Page 7).

Due to variations in TMEM16F expression levels and the ionophore-mobilized

intracellular Ca^{2+} levels, the maximum AnV fluorescent intensities often vary among cells that express the same construct. This variation was also observed in cells expressing different TMEM16F mutations. Based our experience, we did not see any apparent correlation between the total fluorescent change and $t_{1/2(\text{max})}$ values. For instance, the gain-of-function TMEM16F mutants scramble faster than WT TMEM16F, but their final maximum intensity could be anywhere from lower to higher than that of WT. Hence, we could not employ maximum AnV intensity as the final read-out for scrambling activity. This is a common issue for the microscopy-based fluorescent scrambling assays (Yu et al., eLife 2015, DOI: 10.7554/eLife.06901.001).

What does the D703R loss of function mutation do when coupled with F518K or Y563K or (D703R/F518A for that matter)? Based on the authors claims, we would expect that this double mutant would still be a constitutive scramblase.

Reponses: We found that around 34% of F518K-D703R and 93% Y563K-D703R expressing cells exhibited constitutive lipid scrambling activity. By contrast, when coupled to any other gain-of-function mutations of the tested inner gate residues, D703R abolished their spontaneous scrambling activity. This data is now shown in Fig. 5c and 5d of the revised manuscript (previously Fig. 4c and 4d).

The triple mutant was a good idea and it is very active at PS scrambling. Does the Ala triple mutant have to gate?? The authors could have taken the closed, Ca^{++} TMEM16A structure, make the triple mutant (don't force TM helices apart) and look at whether they see permeation. This is important. What is your model telling you about restudies blocking lipid passage versus biasing the channel to different functional states. Ala is interesting too as it is still hydrophobic. I like the bulky residue work to show reduced PS exposure; however, the F518W is still opposite of what might be expected, and the authors haven't addressed whether the bulky residues are pushing the channels to different states (closed versus open) or simply making it impossible for the open state to permeate (unless I missed it). Along these lines the authors say F518L is a LOF, but isn't F bigger and more hydrophobic than L? This confused me some.

Reponses: As shown in Fig. 1d, more than 90% of cells expressing the triple Ala mutations showed constitutive lipid scrambling activities even in the Ca^{2+} -free medium. Based on this observation, we believe that the permeation gate of this TMEM16F triple mutation likely adopts a constitutively open conformation.

The effects of F518W and F518L on scrambling were confusing to us, too. After searching the literature, we found that Trp, despite its larger size compared to that of Phe and Leu, is the most hydrophilic residue among the three amino acids (hydropathy indexes for Trp, Phe and Leu are -0.9, 2.8 and 3.8, respectively) (Kyte, J. and R. F. Doolittle J Mol Biol 157, 105-32 (1982)). Leu, the smallest among the three, on the other hand, has the highest hydrophobicity index. This suggests that both size and hydrophilicity (or hydrophobicity) of the inner gate residues are important in controlling lipid permeation.

As stated on lines 180-181 – all mutations at the inner gate/neck region caused GOF. Interesting, and not unexpected.

Reponses: Agree.

On line 213 page 10, these mutants from the Tien paper that destroy Ca⁺⁺ binding also abolish ion channel activity, right? If so, please mention that. I do like that the authors show that the single point GOF mutants (except two) can't facilitate scrambling when the Ca⁺⁺ site is knocked out suggesting that the GOF mutants can't gate the channel on its own, and possibly hinting that the closed state of the channel can't necessarily be made to flip lipids simply by changing a residue in the pathway. That said, the bulky, aromatic residues F518 and Y563 can flip without Ca⁺⁺ when a lysine is introduced. Are these scramblases still gating like normal or not? This is interesting. Is it the sign of the charge that matters (because the assay is testing PS flipping) or simply the charge? Double mutants with D703R+F518D/E/R and D703R+Y563D/E/R, would be very interesting.

Reponses: We thank the reviewer for the great suggestions.

As we previously showed (Tien et al., eLife 2014), TMEM16A-D734R (corresponding to TMEM16F-D703R) showed the strongest effect on abolishing TMEM16A channel activation. This is the reason why we chose this mutation for TMEM16F in this manuscript. To inform the readers, we added this information in Page 7 "...the Ca²⁺-binding site mutation D703R, whose equivalent mutation in TMEM16A completely eliminated Ca²⁺-dependent activation of the CaCC²⁰, abolished phospholipid scrambling..."

The fact that only positively charged mutations (F518K and Y563K) remain scrambling-competent when coupled with D703R further highlights the importance of charge properties of these mutations on controlling lipid scrambling. As PS is the only major phospholipid that carries a negative charge, we believe that the positive charge at the inner gate could facilitate PS permeation and likely favors PS over the other phospholipids. It will be interesting to test this hypothesis and understand the role of the inner gate residues in controlling scrambling of other phospholipids. Unfortunately, the technical limitations of the functional assays do not now allow us to clearly distinguish permeation, gating and selectivity for the scramblases. Our current imaging assay can only detect PS externalization. To address this important question, we thus hope and aim to develop new assays to measure the permeation of other phospholipid species through TMEM16F in future studies.

Considering the close proximity between the inner activation gate and the Ca²⁺ binding sites (Figs. 1c and 7), it is plausible that these two critical regions are directly coupled. It is likely that Ca²⁺ binding directly opens the inner gate, and disrupting the gate can affect the apparent Ca²⁺ binding affinity. In terms of the charged residues in the inner gate, we totally agree with the reviewer that the lysine mutations of the bulky, aromatic residues F518 and Y563 could make the gate constitutively open without Ca²⁺, as evidence by the strong cytotoxicity from the single mutations and the constitutive activity

of the F518K-D703R and Y563K-D703R in which Ca²⁺ binding is abolished. The detailed mechanism of how the inner activation gate and the Ca²⁺ binding site mutually affect each other requires further investigation. We hope our current identification of the inner activation gate can provide some useful information to facilitate further mechanistic studies on TMEM16 gating.

With regard to the TMEM16A mutations discussed on lines 235-238 please let us know here if L543 and I637 are the homologous sites to F518 and Y563. Also, please indicate what you mean by opening in the text: Cl⁻ channel activity or lipid scrambling?

Reponses: We apologize for the confusion. L543 and I637 of TMEM16A are corresponding to F518 and I612 of TMEM16F, respectively. To avoid confusing the readers, we have made a lookup table in Supplementary Fig. 1d for the corresponding gate residues in TMEM16F, TMEM16A and nhTMEM16.

Nowhere in the paper can I find the word “serine” except in the references. On line 138, the authors discuss PS exposure – what the assay is detecting – but it isn’t discussed in detail. The simulations probe PC movement, but PS is the lipid of interesting. There is a lot of difference in the lipid types when it comes to gating, selectivity, and function (the Accardi group has nicely shown this in several papers). The authors should clearly point this out so that the readers appreciate this.

Reponses: We really appreciate the reviewer for pointing this out. The abbreviation of PS is now explained in the revised manuscript (1st sentence of the manuscript, Page 2). We have also added a new section in the main text detailing how we measured lipid scrambling, including the description using Annexin V binding to the externalized PS as an indicator of scrambling activity (Page 7). More details were included in the Methods section, as well.

Minor Points

The authors state that the SE site charged residues are not conserved – this isn’t true is it? They may have a charge reversal, but they exist in almost all family members, right?

Reponses: We really appreciate this question. Please see our response to ‘Simulation-Major concern #1’ (Pages 3-4) for details.

Figure 5 caption mentions WT in panel A, but I don’t think WT is shown.

Reponses: We apologize for missing the WT data. It is now included in Fig. 6a.

On lines 278-279 the authors state that the motions are centered on TM6. However, doesn’t it appear that TM4 and TM3 are just as big when we compare different solved structures from different subfamilies?

Reponses: The statement is based on the prevailing Ca^{2+} -dependent gating model in which Ca^{2+} binding to the binding sites triggers a large conformational change of TM6 and subsequently opens TMEM16 proteins (Paulino et al., 2017, Nature). We totally agree with the reviewer that TM3 and TM4 may also undergo large conformational changes and could contribute to channel gating. To avoid future confusion, we have changed this sentence to “Upon Ca^{2+} binding, large-scale conformational changes, likely including the movement of TM6 (Supplementary Fig. 1b) around a conserved glycine hinge^{25,30} and the rearrangement of TMs 3-5, separate TM4 and TM6 (Fig. 1c).”

Reviewer #3 (Remarks to the Author):

In this report, the authors introduced mutations in several amino acids of mouse TMEM16F that functions as a phospholipid scramblase, and propose a model how this protein transports phospholipids between the two leaflets of plasma membranes. I appreciate the effort to characterize this interesting membrane protein. But unfortunately, the data are preliminary, and are not well performed. My comments are as follows.

1. The scramblase activity (the exposure of phosphatidylserine, PS) of the TMEM16F derivatives was assayed by transient-over expression in 293T cells. The authors often observed caspase-positive cells in the Ca^{2+} -ionophore treated transfectants. This is not a good experiment, or it is a dirty experiment. As the authors are aware, PS is exposed in apoptotic or necrotic cells. The authors argue that they examined the PS-exposure in caspase-negative cells. But, we do not know how much caspase should be activated to induce PS-exposure, and how the caspase-detecting system is sensitive. To study the PS-exposure by TMEM16 family member, it is essential to study with the stable transformants. Please see reference 5, and Gyobu et al. (PNAS 114, 6274, 2017).

Reponses: We respectfully disagree with the reviewer’s comments above.

First, transient transfection is a prevailing method to overexpress exogenous proteins. In addition, X-tremeGENE™ 9 DNA Transfection Reagent used in this study is known to have “extremely low cytotoxicity for maximum post-transfection cell viability” (<https://www.sigmaaldrich.com/catalog/product/roche/xtg9ro?lang=en®ion=US>). Comparing to stable transformation via viral infection, transient transfection is much more convenient, especially when testing with large number of mutations. In fact, overexpression of TMEM16 lipid scramblases using transient transfection have been used by other labs [Di Zanni et al., Cell Mol Life Sci, 2018; Whitlock et al., J Gen Physiol, 2018; Yu et al., Elife, 2015; Jiang et al., Elife, 2017]. To experimentally address the reviewer’s concern, we also examined the scrambling activity of the HEK293T cells stably expressed TMEM16F. Our results showed that there is no significant difference in scrambling activity between the stably- and transiently-expressed cells (Response Fig. 2a, see below).

Second, to further demonstrate that no obvious apoptosis or necrosis was induced by transient transfection, we conducted the following additional experiments.

- 1) We used a caspase activity dye (TF3-DEVD-FMK) to detect active caspase in the TMEM16F transiently transfected cells. As shown in Response Figure 2b, the active caspase levels in the TMEM16F-transfected cells and the non-transfected control cells are indistinguishable. Moreover, the TMEM16F-transfected cells that showed robust scrambling exhibited minimal active caspase signal 10 minutes after ionomycin treatment (Response Fig. 2b). Together, these data indicate that neither transient transfection nor overexpression of TMEM16F induces apoptosis or necrosis.
- 2) We treated the TMEM16F-transfected cells with 10 μ M pan-caspase inhibitor (Q-VD-OPh) and measured ionomycin-induced lipid scrambling (Response Fig. 2c). We found that the Q-VD-OPh-treated and non-treated control cells had similar scrambling activities. This experiment suggests that within the 10-minute window post-ionomycin treatment, CaPLSase activity but not caspase activity is responsible for the lipid scrambling observed in the TMEM16F-transfected cells.

Third, we would like to clarify that we did not observe significant caspase signal in the Ca²⁺-ionophore treated HEK293 cells that were transiently transfected with TMEM16F plasmids (Figs. 3c, 5d, 6e, S3, S5a and Response Fig. 2b). The only cases where we found strong caspase activities were Y563A in regular Ca²⁺-containing medium (Fig. 3c), F518K and Y563K in Ca free medium (Supplementary Fig. 5a). In these instances, no ionomycin was used. Thus, we suggest that the caspase activities were due to the strong spontaneous gain-of-function mutational effects, which induced cytotoxicity and cell death.

In summary, our results support that transient transfection is a reliable and efficient method to heterologously express CaPLSases without introducing artifacts from apoptosis.

Response Figure 2. Transient transfection of TMEM16F is suitable to characterize CaPLSase activity in HEK293 cells. **a**, Transiently and stably expressed TMEM16F HEK293T cells show no

difference in scrambling activity as quantified by $t_{1/2(lmax)}$. **b**, Heterologous expression of TMEM16F does not induce cytotoxicity. Representative images of TMEM16F-transfected HEK293 KO cells (purple) before (top), 5-minutes (middle) and 10-minute (bottom) after ionomycin treatment. CF 640R-tagged AnV (AnV, red) and TF3-DEVD-FMK (Casp, green) label surface-exposed PS and active caspases, respectively. Note that no difference on caspase activity was observed in TMEM16F-transfected and non-transfected cells. **c**, Inhibiting caspase activity does not affect TMEM16F phospholipid scrambling. TMEM16F was transiently expressed in TMEM16F-knockout (KO) HEK293T cells. After 5h of transfection, cells were treated with 10 μ M Q-VD-OPh caspase inhibitor overnight. The caspase inhibitor was also included in the scrambling assay. Statistical analysis in a and c was performed using unpaired two-sided Student's *t* test. n.s, not significant. Error bars indicate SEM.

2. With the data obtained with the mutants, the authors discuss the detailed scrambling mechanism, and propose that Ca²⁺ causes the conformational change from Open to Closed structure (Figure 6). Recently, Bushell et al (<http://dx.doi.org/10.1101/447417>) determined the tertiary structure of Ca²⁺-bound and unbound forms of TMEM16K, a homolog of TMEM16F that scrambles phospholipids in Ca²⁺-dependent manner. They report that both Open and Closed conformation can be found in Ca²⁺-bound protein. Thus, the proposal in this manuscript is mis-leading.

Reponses: We respectfully disagree with this comment based on the following reasons.

First, despite the significant progresses in recent structural characterizations of TMEM16 proteins, the current understanding on how the calcium-activated lipid scramblases work is still primitive. While we agree that the Biorxiv manuscript of TMEM16K provides new insights on the structure-function relationship of TMEM16 scramblases, we disagree with the reviewer's comment that our study is "mis-leading." Our extensive experimental and computational data reveal an inner gate that needs to undergo conformational transition to mediate scrambling in TMEM16F CaPLSase. Importantly, our findings and proposed model are consistent with the well-accepted calcium-dependent 'gating' concept of TMEM16 proteins, in which the opening and closing of the substrate permeation pathway are tightly controlled by Ca²⁺ binding.

Second, the statement of "both Open and Closed conformation can be found in Ca²⁺-bound protein" based on Bushell et al. (<http://dx.doi.org/10.1101/447417>) needs further scrutiny. Indeed, the Ca²⁺-bound TMEM16K X-ray structure was solved in the presence of rather extremely high 100 mM Ca²⁺; while the second cryo-EM structure was determined in the presence of 500 nM Ca²⁺ and very high 10 mM EGTA, which was referred to as "nominal Ca²⁺-free conditions" by the authors. In the cryo-EM structure of TMEM16K (in 500 nM Ca²⁺ and 10 mM EGTA), although the authors observed electron density at the Ca²⁺ binding site that they tentatively assigned as Ca²⁺ ions, we believe this structure may not truly represent a fully Ca²⁺-bound state given the extremely low presence of Ca²⁺ in their preparation. Instead, it is more likely that the TMEM16K cryo-EM structure resolved in 500 nM Ca²⁺ represents an intermediate or transition state on its trajectory to an open or a closed conformation of TMEM16K. In fact, this 500nM Ca²⁺-bound TMEM16K structure superimposes very well with the cryo-EM structure of the Ca²⁺-free TMEM16K structure with minimal conformational changes (Fig. 6 from Bushell et al). These observations suggest that the open TMEM16K conformation in the

presence of 100 mM Ca^{2+} captured by X-ray crystallography likely represents the open scramblase whereas the 500 nM Ca^{2+} and Ca^{2+} -free TMEM16K cryo-EM structures more likely represent the closed scramblase.

Third, our TMEM16F scramblase gating model is actually consistent with the recent structural observations and computational predictions. In Bushell et al. (<http://dx.doi.org/10.1101/447417>) manuscript, the authors described that “The hydrophobic neck is lined with residues from TM4-7 (Tyr366, Ala367, Leu416, Ser415, Thr435, Leu436 and Tyr507), which together form a physical barrier to lipid movement” in their “ Ca^{2+} -bound, but closed state”. Residues Ala367 and Leu436 are actually equivalent to the activation gate residues Phe518 and Ile612, respectively) identified in our current study. Although TMEM16F Y563 is not equivalent to the hTMEM16K residues listed above, it is equivalent to hTMEM16K Cys412 residue, which is one turn below Leu415 and presumably should face toward the groove.

Taken together, the recent structural findings actually lend further support to our experimental results and the gating model.

Reviewer #4 (Remarks to the Author):

Review Manuscript by Le et al. entitled “An inner activation gate controls TMEM16F phospholipid scrambling”

In this manuscript Le et al. investigate the lipid permeation pathway in the Ca-activated scramblase TMEM16F by using mutagenesis, scramblase activity assays, electrophysiology and MD simulations.

As template to guide the modeling and the selection of the residues to be mutated, they used the structures of Ca-bound and Ca-free mouse TMEM16A and of the Ca-bound fungal nhTMEM16.

The Authors obtained two main results.

First they identified as inner gate for lipid transport three hydrophobic residues, F518, Y563 and I612, in the middle of the phospholipid permeation pathway. Interestingly a single mutation of one of these aminoacids can disrupt the gate, in some cases leading to a constitutively active scramblase.

Second they converted the Ca-activated chloride channel TMEM16A into a constitutively active scramblase by a single mutation of one of the corresponding aminoacids (L543K) identified in TMEM16F. This second result is really surprisingly.

The overall similarity of all TMEM16 is not a surprise and Hartzell's and Nagata's groups (Yu et al 2015, Gyobu et al 2016, Gyobu et al 2017) converted TMEM16A into a scramblase, but by introducing a stretch of 35 aminoacids of a TMEM16 scramblase into TMEM16A!

The results are interesting, but more experiments are needed to be completely convincing.

Reponses: We thank and appreciate the reviewer for the positive comments and constructive suggestions. Detailed responses are listed below.

Major

1) In most TMEM16s scrambling activity has been associated to little selective ionic current (TMEM16F Yang et al 2012; TMEM16E Dizanni et al 2018; nhTMEM16 Lee et al 2016; afTMEM Malvezzi et al 2013). It makes exception the recently identified Dictyostelium discoideum TMEM16 (Pelz et al 2018), but in this case very limited data are available. Additionally, gain of function mutations in scrambling activity have been associated to gain of function current activity both in TMEM16E (see Fig. 6 Di Zanni et al 2018) and TMEM16F (D409G, see Fig9 Scudieri et al, J Physiol 2015). Therefore comes up very naturally the question “what happens to the current of TMEM16F with the mutations described in this manuscript?” Are the gain of function mutations for scrambling activity of TMEM16F also showing a gain of function in current activity? I do not ask for a full characterization of all mutations, but I think it will complete the work to show whether there is a gain of function activity at least for the most prominent ones. Possibly use TMEM16F-KO HEK cells.

Reponses: We really appreciate this insightful question. Due to the challenges (mainly due to cytotoxicity and change of cell adhesion) on recording the strong gain-of-function TMEM16F mutations, we focused on Y563A, a milder gain-of-function mutation, and systematically characterized its channel biophysical properties (included in Supplementary Fig. 4). Consistent with the observed dramatic changes in lipid scrambling, channel activation, ion selectivity and rundown are all altered in Y563A-TMEM16F channel. Compared to WT TMEM16, this mutant channel is more sensitive to calcium and voltage (Supplementary Fig. 4), has an increase in P_{Na}/P_{Cl} (Response Fig. 3c-f), and for an unclear reason, abolishes current rundown, which is pronounced in WT TMEM16F (Response Fig. 3a,b). These experiments suggest that the putative activation gate not only controls lipid permeation but also likely regulates ion transport. The gain-of-function of channel and scramblase activities of Y563A thus further support our conclusion that the residues identified in this study play important roles in gating TMEM16F. We now include the characterization of Y563A channel activation in Supplementary Fig. 4 in the revised manuscript. The mechanisms of how the inner gate residues control ion selectivity and rundown need further examination for better understanding. To avoid confusing the reader, we show these characterizations below for the reviewer's reference.

Response Figure 3. Functional characterization of TMEM16F WT and Y563A's channel rundown and ion selectivity. **a,b**, Whereas TMEM16F WT exhibited pronounced rundown under presence of 100 μM intracellular Ca^{2+} (**a**), Y563A abolished channel rundown (**b**). **c,d**, Measurements of the reversal potentials (E_{rev}) for TMEM16F WT (**c**) and Y563A (**d**). Black traces denote currents at symmetric 140 mM NaCl; red traces denote currents upon switching to an intracellular solution with low 14 mM NaCl. A voltage ramp ranging from -120 mV to +120 mV was used to elicit channel activation and followed by a reserved +120 mV to -120 mV ramp, which was used to measure reversal potentials. Currents were recorded under 100 μM intracellular Ca^{2+} . **e,f**, Changes in the reversal potential (E_{rev}) of TMEM16F WT and Y563A (**e**) and their permeability ratio $P_{\text{Na}}/P_{\text{Cl}}$ (**f**). Two-tailed un-paired Student's *t* test: *p*-values are <0.0001 in **e** and **f**.

2) Page 7 A cytotoxic effect due to constitutive scrambling activity in cells expressing mutant TMEM16F is described. It is probably supporting your data the paper by Dizanni et al 2018 where it is reported a rounded cell morphology when it is expressed a gain of function mutant of TMEM16E. Another gain of function mutation has been previously described for TMEM16F (D409G) and used in this work. Have you observed also here a cytotoxic effect? Reference for D409G mutant is missing.

Reponses: We thank the reviewer for reminding us the paper by Dizanni et al 2018 about the cytotoxic effect of TMEM16E GOF mutation. We have included this reference to our revised manuscript. We have also cited Suzuki, 2010 Nature paper, which identified the gain-of-function about D409G.

Based on our caspase and Annexin V staining for D409G (Response Fig. 4), we did not observe obvious cytotoxic effect or spontaneous PS exposure in the D409G transfected HEK293 cells. This is consistent with our observation in the revised main Fig. 2a (0 min). Our observations are different from the previous report by Segawa et al., PNAS

2011, where the authors showed that the D409G-overexpressing Ba/Fe or W3-Ildm cells exhibited constitutive exposure of PS on their cell surfaces. We postulate that the discrepancy is likely due to the differences in the cell lines used in the two studies. The potentially more active calcium dynamics in the immune cell lines might be sufficient to trigger TMEM16F-D409G scrambling; whereas the basal calcium level in HEK293 cells is insufficient to activate D409G. Other alternative explanations could be that 1) TMEM16F might have different post-translational modifications in two different cell lines, 2) the TMEM16F construct used in the Segawa PNAS paper differs in splicing isoform or species from our mouse TMEM16F construct. As the molecular mechanism of how D409G leads to gain-of-function phenotype is still unknown, the exact reason of this discrepancy needs future investigation and is beyond the scope of this study.

Response Figure 4. TMEM16F-D409G does not spontaneously scramble phospholipids when expressed in TMEM16F knockout HEK293T cells. No obvious cytotoxic effect was observed. Representative images of D409G-expressed cells (green) without ionomycin stimulation. CF 640R-tagged AnV (red) and TF3-DEVD-FMK (purple) label surface-exposed PS and active caspases, respectively.

3) The conversion of TMEM16A CaCC to a scramblase with a single mutation is a key point. I think that further evidence should be given to support this surprising result
a. Please show in Figure 5 also cells expressing TMEM16A-L543K showing cytotoxicity.

Reponses: TMEM16A-L543K cytotoxicity is shown in Response Fig. 5 (see below). Please note that the cytotoxicity can be largely prevented when the transfected cells were cultured in Ca^{2+} -free medium (previously Fig. 5d, now new Fig. 6d).

Response Figure 5. TMEM16A-L543K needs to be expressed in a Ca^{2+} -free medium to suppress its cytotoxicity. Representative images of TMEM16A L543K-expressing the cells (green) cultured in Ca^{2+} -containing and Ca^{2+} -free media without ionomycin stimulation. CF 640R-tagged-AnV (purple) labels PS positive cells. White asterisk labels apoptotic cells with positive AnV (red) and strong cytosolic TF3-DEVD-FMK (indicative of cleaved caspases 3/7) staining.

b. Is An-V assay performed in TMEM16F KO cells? (specify in the legend) If not it should be done.

Reponses: Yes, all scrambling assays were done in the TMEM16F-KO HEK293T cells. A description was added in each figure legend.

c. Can scramblase activity of TMEM16A L543K blocked by TMEM16A blockers such Ani9, MONNA or CaCCinh01?

Reponses: Yes, Ani9 can inhibit the scrambling activity of TMEM16A-L543K. The data has been added to the revised Fig. 6d. MONNA and CaCCinh01 have not been tested.

d. Do in excised patches currents mediated by TMEM16A-L543K run down as WT?

Reponses: TMEM16A-L543K showed faster rundown than WT (Response Fig. 6). The exact reason why L543K accelerates channel rundown/desensitization is unknown. According to our recent effort in understanding TMEM16A rundown, L543K may destabilize the pore and accelerate the collapse of the ion permeation pore, which we believe is the structural basis of TMEM16A rundown.

Response Figure 6. Functional characterization of TMEM16A WT, L543K and I637F's channel rundown. $t_{1/2}$ is the time needed for TMEM16A channels to desensitize to 50% of its original current amplitude. The smaller the $t_{1/2}$ value, the faster the channel desensitization. One-way ANOVA with Tukey's multiple comparisons test: p-values are 0.0023 for L543K and 0.2533 for I637K.

e. Fig 5a. Why there are not tail currents upon repolarization? Do you see them with WT (with Ca)?

Reponses: Membrane depolarization alone fails to activate TMEM16A WT in the absence of Ca^{2+} (Fig. 6a). Nevertheless, under submicromolar Ca^{2+} , TMEM16A WT channels deactivate slowly when the membrane is repolarized, a hallmark of CaCC.

Different from WT, the gain-of-function TMEM16A- L543K and I647K channels can be activated by voltage alone in the absence of Ca^{2+} . The lack of obvious tail current upon membrane repolarization for L543K channels is likely due to its low open probability and ultra-fast channel deactivation, which leads to instantaneous channel closure. This observation is consistent with the TMEM16A Q645A gain-of-function mutation (Peters et al., *Neuron* 2018), which also exhibits no tail current due to instantaneous deactivation. The lack of tail current in these gain-of-function mutations thus suggests that voltage alone has limited contributions to channel activation when Ca^{2+} is absent.

In symmetric chloride I would expect an inward current upon repolarization. Is selectivity Cl/cations affected for L543K mutant? In Yu et al 2015 chimeric constructs TMEM16A/TMEM16F show a change in selectivity.

Yes. TMEM16A- L543K indeed has altered ion selectivity as shown below (Response Fig. 7).

Response Figure 7. L543K mutation altered the ion selectivity of TMEM16A. a,b, Measurements of ion selectivity of TMEM16A WT (a), L543K (b). Black traces denote currents at symmetric 140 mM NaCl. Red traces denote currents upon switching to an intracellular solution with low 14 mM NaCl. A voltage ramp ranging from -100 mV to +100 mV was used to elicit channel activation. Currents were recorded under 100 μM intracellular Ca^{2+} . c, The permeability ratios $P_{\text{Na}}/P_{\text{Cl}}$ of TMEM16A- WT and -L543K. Two-tailed un-paired Student's *t* test: p-value is 0.0191.

f. TMEM16A mutant I637K does not show scrambling activity, but only current activity. It should be discussed. Is mutant S588K showing current in high Ca^{2+} ?

Reponses: A discussion about I637K is now added into the main text (Page 13, last paragraph).

Although S588K cannot be opened by voltage alone, it indeed can be activated by high Ca^{2+} as shown below (Response Fig. 8).

Response Figure 8. TMEM16F S588K cannot be activate without Ca²⁺. Representative recording showing activation of S588K when 0 Ca²⁺ (black trace) or 100 M Ca²⁺ was applied to the cytosolic side of an inside-out patch excised from HEK293T cells expressing TMEM16A S588K.

4) Student's test was used, however to compare multiple data is not appropriate. Anova and an appropriate post-hoc test should be used.

Reponses: We thank the reviewer for pointing this out. We have switched to one-way ANOVA (Turkey's multiple comparison test) when comparing multiple data groups. In the cases where only two data groups were involved, we used Student's t test.

Medium

1. It is missing in the main manuscript a figure with a representative experiment for scrambling activity of wild type and a gain of function mutant TMEM16F. Reference to supplementary figure and to another paper of the same authors is given (submitted to JBC), however it would be better to have in the main text at least a graph of total fluorescence vs time, from which you estimate $t_{1/2}$, to appreciate the validity of your analysis on TMEM16F mutants (as in Fig4 of the submitted paper and in Supplementary figure 4). Additionally, the value of the $t_{1/2}$ includes the wash-in of the solution and by the time necessary for enough annexin-V binding to PtdSer. It should be evaluated. For example from the D409G gain of function mutation, that scrambles without ionophore (in the supporting materials also ionomycin is applied) or from apoptotic cells.

Reponses: We thank the reviewer for the suggestion. We have now moved original Supplementary Fig. 4 to the main text as the new main Fig. 2, in which we showed the examples of WT, the known gain-of-function and loss-of-function TMEM16F scramblases, and our analysis.

The reviewer is absolutely correct that the $t_{1/2(l_{max})}$ value is only an estimation of lipid scrambling activity. $t_{1/2(l_{max})}$ includes the time that is needed for solution exchange and also the time necessary for Annexin V diffusion and binding. Considering the lower sensitivity of the epifluorescence microscopy and the above-mentioned complications,

we believe the real lipid scrambling speed should be faster than the $t_{1/2(I_{max})}$ values measured. We are fully aware of the limitations of the microscope-based methods. Therefore, in this study, we focused more on the strong gain-of-function mutations especially those that have constitutive scrambling activities. When we compared the $t_{1/2(I_{max})}$ values, we always checked their spontaneous activities to ensure that our conclusions were not just based on the $t_{1/2(I_{max})}$ values.

To avoid potential complications, all the PS-positive cells that either underwent apoptosis or showed spontaneous scrambling were excluded from our $t_{1/2(I_{max})}$ measurement. We clarified this in the main text and also the Methods section.

As we discussed in our response to Major concern #2, D409G is a much milder gain-of-function mutation compared to F518K and Y563K. D409G induces neither spontaneous scrambling nor cytotoxicity.

2. It should be specified that Annexin V binding assay and ionomycin application is made in presence of Ca even when cells have been kept in Ca-free medium to avoid cytotoxicity (for example page 11).

Reponses: We thank the reviewer for the suggestion. We have specified it in the methods, main text and legends.

3. Page 12 Electrophysiology experiments are not properly described in the results

Reponses: More detailed description about electrophysiology is now included in Page 13-14.

4. Specify in each legend if TMEM16F KO cells have been used or HEK cells

Reponses: Done. All of our phospholipid scrambling experiments were done using TMEM16F KO cells. Electrophysiology recordings of TMEM16F WT and mutants were performed in TMEM16F KO cells whereas recordings of TMEM16A WT and mutants were performed in normal HEK293 cells.

Minor

I do not find necessary to use acronyms for gain of function and loss of function (as pure suggestion I would not use GOF and LOF).

Reponses: The acronyms have been removed.

CaPLSase stands for Ca-activated phospholipid scramblase. Therefore authors should pay attention whether to use PLSase instead of CaPLSase. For example in the text they often report constitutively active CaPLSase, shouldn't it be PLSase?. In some mutants the Ca-dependence is actually lost.

Reponses: Our intention to use CaPLSase was to avoid confusion with other PLSases such as xkr8 and rhodopsins. In addition, many constitutively active TMEM16F mutations are likely still calcium-sensitive (removal of extracellular calcium or combination of the calcium binding site mutation can reduce the gain-of-function phenotype). We believe that the nomenclature of CaPLSase in the context of this manuscript would be suitable. Nevertheless, we hope the field will come up with a better nomenclature for these special proteins in the future.

Page 3 line 44. PGD is introduced but never used in the text

Reponses: The term PGD has been removed from the text.

Page 4 line 73, 14 line 294 CaCC channel .

Reponses: Thanks. Changed to 'CaCC'.

Page 6 Add somewhere that you used an annexin-V based assay.

Reponses: We added a new section "Application of an optimized Annexin V (AnV)-based imaging assay to quantify CaPLSase activity" in the Results to explain the details of this assay.

Page 7 line 138. Specify that the other mutants do not show activity in absence of ionophore.

Done.

Page 13 line 268, specify that corresponds to F518 in TMEM16F, as reported later line 293

Done.

Pag 13,line 280 groove instead of grove

Corrected.

Page 13,line 282, add ref to the 'credit card mechanism'

Done.

Pag 18 line 384 manually

Corrected.

Supplementary figures are referred to as Supplementary/Supplemental Fig 2 or Supplementary Fig S2

Thanks. Changed according to the instruction of NC.

Figures

Legends in figure panels are too small

Corrected.

Fig 2b,c, fig3b, Fig4b, fig 5e Reduce the number of ticks with numbers in the Y axis
Done.

Fig 2b Xaxis labels are hidden by the panel below and the right part is missing
Corrected.

Fig3b, 5e histogram is missing for some mutations. If it is done because too few cells scramble, it should be written in the legend.
Figures and legends are revised accordingly.

Fig 4b For the constitutively activated TMEM16F mutants that caused >80% of spontaneous PS exposure add at least the pie graph.
This panel (current Fig. 5b) shows $t_{1/2(I_{max})}$ measurement. Since all the strong gain-of-function mutations showed spontaneous scrambling activity, almost all the expressing cells were PS positive. This made $t_{1/2(I_{max})}$ measurement impossible for these mutations. Instead of adding pie graphs, we added '##' to mark these mutations with strong spontaneous activities.

Fig 5 time and current scale is missing.
Added.

Fig5b increase height of the panel
Done.

Fig S2a letters are too small (and stretched). Difficult to read. Black is too faint (at video)
Corrected.

Legends

Fig1 line 632 TMEM16F
Corrected.

Fig4b Although cells were kept ion Ca-free medium, when ionomycin is applied Ca is present. Specify it in the legend
Thanks for pointing this out. This is important. We have added the experimental detail in the legend and also described it in the Methods section.

Fig 5, specify when ionomycin was used.
Done.

Fig 5d specify that representative images of L543K-TMEM16A-expressing cells are from cells kept in Ca-free medium-
Done.

In some panels is missing the number of experiments. I would add it to the legend.

Added.

Fig. S8 I assume that legend is wrong. Specify that at $t=0\text{min}$ there is no annexin (otherwise I would expect staining of L543K mutant)

Reponses: ~40% of L543K-expressing cells are PS positive without stimulation. For the example shown in Supplementary Fig. 7, we chose the cells that did not show spontaneous PS exposure. Therefore, at $t=0\text{min}$, the L543K positive cells were Annexin V negative. This has been clearly noted in the legend.

Reviewers' Comments:

Reviewer #2:

Remarks to the Author:

This manuscript is much more clear to me now, and I think it is better presented. I have two comments.

1. Lily Jan's lab recently showed that I612 in TMEM16F also influenced lipid scrambling and vesiculation in stable lines (Han et al. PNAS 2019). I think that this work confirms some of the results in the present paper, and also make me feel less worried about the transient transfection versus stable transfection. That said, I think that the Han paper should be cited here.

2. I am not sure how the authors did the error in the MD simulation energy profiles, but I expect the errors are being suppressed. The authors should make sure that they compute autocorrelation times for the particles in different windows along the pathway to subsample the MD data used in Figure 1 for the error analysis. Don't use all the collected data, because your frame rates are certainly going to be correlated. Typically WHAM computations handle this properly, and pyMBAR will calculate the correlation times for you or the GROMACS WHAM plugin will too. In this case, your analysis is like doing a WHAM with no umbrella.

Reviewer #3:

None

Reviewer #4:

Remarks to the Author:

Review Manuscript by Le et al. entitled "An inner activation gate controls TMEM16F phospholipid scrambling"

The revised version of the Manuscript is strongly improved and the experiments I requested have been performed

I have few minor points:

Methods section: intracellular solution with 100 microM Ca is missing

Page 2 line 9 of the main text change "shown to involve in blood coagulation" into to be involved
Fig S1, panel 1 The height of the rows in the alignment is insufficient and the letters are partially cut. Correct nhTMEM16 instead of nhMTM

Fig s7, panel b Ca²⁺, ²⁺ superscript; panel f add a space between WT and L543K in the x-axis label.

REVIEWERS' COMMENTS:

Reviewer #2 (Remarks to the Author):

This manuscript is much more clear to me now, and I think it is better presented. I have two comments.

We sincerely appreciate the reviewer's help in improving our manuscript. The responses to the comments are listed below.

1. Lily Jan's lab recently showed that I612 in TMEM16F also influenced lipid scrambling and vesiculation in stable lines (Han et al. PNAS 2019). I think that this work confirms some of the results in the present paper, and also make me feel less worried about the transient transfection versus stable transfection. That said, I think that the Han paper should be cited here.

We appreciate the reviewer for reminding us the Han et al. PNAS 2019 paper. The PNAS paper is now cited in Page 9 of the manuscript.

2. I am not sure how the authors did the error in the MD simulation energy profiles, but I expect the errors are being suppressed. The authors should make sure that they compute autocorrelation times for the particles in different windows along the pathway to subsample the MD data used in Figure 1 for the error analysis. Don't use all the collected data, because your frame rates are certainly going to be correlated. Typically WHAM computations handle this properly, and pyMBAR will calculate the correlation times for you or the GROMACS WHAM plugin will too. In this case, your analysis is like doing a WHAM with no umbrella.

We estimated the standard error of the calculated averages from three independent simulations. WHAM and MBAR analysis were not used. A brief description about how we estimate the error has been added in Fig. 1d legend and also the Methods section (Page 28).

Reviewer #4 (Remarks to the Author):

Review Manuscript by Le et al. entitled "An inner activation gate controls TMEM16F phospholipid scrambling"

The revised version of the Manuscript is strongly improved and the experiments I requested have been performed

We sincerely appreciate the reviewer's help in improving our manuscript.

I have few minor points:

Methods section: intracellular solution with 100 microM Ca is missing

More details about electrophysiology including the 100 microM Ca are included in the Methods section.

Page 2 line 9 of the main text change "shown to involve in blood coagulation" into to be involved

Corrected.

Fig S1, panel 1 The height of the rows in the alignment is insufficient as the letters are partially cut.

Corrected.

Correct nhTMEM16 instead of nhMTM

Corrected.

Fig s7, panel b Ca²⁺, 2+ superscript; panel f add a space between WT and L543K in the x-axis label.

Changed.